# Development and validation of HIV SMRTcap for the characterization of HIV-1 reservoirs across tissues and subtypes

Ghazal Sadri[1☙], Steven T. Nadakal[1☙], William Lauer[1], Justin Kos[1], Parmit K. Singh[2,3], Erin Elliott[1], Catherine W. Kaiser[1], Easton E. Ford[1], Nadia Richardson[1], Kaitlyn M. Shields[1], Elizabeth Hudson[1], Noemi L. Linden[4,5,6], Ali Danesh[5,6], James Powell[7], Peter Warburton[7], Juan Soto[7], Matthew Emery[7], Gintaras Deikus[7,8], Guinevere Q. Lee[4], Susanna L. Lamers[9], Steven J. Reynolds[10,11,12], Ronald Galiwango[10], Jessica L. Prodger[13,14], Stephen Tomusange[10], Taddeo Kityamuweesi[10], Tina Han[15], R. Brad Jones[5,6], Aaron A. R. Tobian[16], Alan N. Engelman[2,3], Robert Sebra[7,8], Susan Morgello[17], Andrew D. Redd[11,12,18], David Sachs[7,8☙], Eric Rouchka[1,19☙], Melissa L. Smith ● [1☙]*

1 Department of Biochemistry and Molecular Genetics, University of Louisville, Louisville, Kentucky, United States of America, 2 Department of Cancer Immunology and Virology, Dana-Farber Cancer Institute, Boston, Massachusetts, United States of America, 3 Department of Medicine, Harvard Medical School, Boston, Massachusetts, United States of America, 4 Department of Medicine, Division of Infectious Diseases, Weill Cornell Medicine, New York, New York, United States of America, 5 Department of Microbiology and Immunology, Weill Cornell Medicine, New York, New York, United States of America, 6 Weill Cornell Medicine, Graduate Program in Immunology and Microbial Pathogenesis, Weill Cornell Graduate School of Medical Sciences, New York, New York, United States of America, 7 Department of Genetics and Genomic Sciences, Icahn School of Medicine at Mount Sinai, New York, New York, United States of America, 8 Icahn Genomics Institute, Icahn School of Medicine at Mount Sinai, New York, New York, United States of America, 9 BioinfoExperts, LLC., Thibodaux, Louisiana, United States of America, 10 Rakai Health Sciences Program, Kalisizo, Uganda, 11 Laboratory of Immunoregulation, Division of Intramural Research, National Institute of Allergy and Infectious Diseases, National Institutes of Health, Baltimore, Maryland, United States of America, 12 Department of Medicine, Johns Hopkins University, Baltimore, Maryland, United States of America, 13 Department of Microbiology and Immunology, Schulich School of Medicine and Dentistry, Western University, London, Ontario, Canada, 14 Department of Epidemiology and Biostatistics, Schulich School of Medicine and Dentistry, Western University, London, Ontario, Canada, 15 Twist Bioscience Corp., San Francisco, California, United States of America, 16 Department of Pathology, Johns Hopkins University, Baltimore, Maryland, United States of America, 17 Division of Neuro-Infectious Diseases, Department of Neurology, Icahn School of Medicine at Mount Sinai, New York, New York, United States of America, 18 Institute of Infectious Disease and Molecular Medicine, University of Cape Town, Cape Town, South Africa, 19 Kentucky Biomedical Research Infrastructure Network Data Science Core, University of Louisville, Louisville, Kentucky, United States of America

☙ These authors contributed equally to this work.

* ml.smith@louisville.edu

## Abstract

Human Immunodeficiency Virus type 1 (HIV-1) is responsible for the global HIV/AIDS epidemic and the establishment of an integrated HIV-1 reservoir remains the primary obstacle to cure. Upon therapy interruption, reactivation of the persistent HIV-1 reservoir propagates viral rebound and mediates continued immunological decline. While furthering understanding of the HIV-1 reservoir is essential for HIV-1 cure, commonly used sequencing strategies are often limited by the reliance on short-read

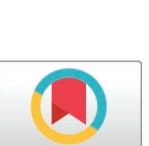

**Data availability statement:** Data and computational tool(s) availability All sequencing data described here, including total HiFi reads per sample and the post-filtered total HIV-containing reads per sample, have been deposited into the Sequence Read Archive and can be retrieved from BioProjects PRJNA1251704 and PRJNA1252688. Custom computational tools are available through GitHub at URL: https://github.com/SmithLabLouisville/SMRTCap.

**Funding:** This work was supported by grants from the National Institutes of Health, including S10OD034432 (MLS), R01DK131926 (MLS, AT), R21MH122368 (MLS), P20GM103436 (ECR), R01AI052014 (PS, ANE), U54AI170791 (ANE), R37 AI181626 (RBJ, NLL, AD), UM1AI164565 (GL, RBJ, NLL, AD, MLS, RG, TK, SJR, ST), R01 AI162221 (GQL), and U24MH100931 (SM). The funders had no role in study design, data collection and analysis, decision to publish, or preparation of the manuscript.

**Competing interests:** We have read the journal's policy, and the authors of this manuscript have the following competing interests: MLS is co-founder and Chief Executive Officer (CEO) of Clareo Biosciences, Inc.; however, the work presented here is unrelated to Clareo Biosciences. RPS is co-founder and CEO of Panacent Bio, Inc.; however, the work presented here is unrelated to Panacent Bio. TH is a full-time employee of Twist Biosciences, Corp. The remainder of the authors have declared that no competing interests exist.

sequencing across separate assays to determine integration sites and proviral integrity – something that does not always adequately resolve complex human genomic repeats or low complexity regions. Simultaneous identification of proviral integration sites and proviral integrity at the single molecule level would enable HIV-1 reservoir characterization with minimal imputation or bioinformatic reconstruction. Here we present HIV Single Molecule Real Time Capture (HIV SMRTcap), a novel molecular and computational pipeline that directly and simultaneously identifies HIV-1 integration sites, defines proviral integrity, and characterizes clonal expansion of HIV-1 provirus-containing cells with single molecule resolution. In combination with long-read, single-molecule, real-time (SMRT) sequencing and custom analytic pipelines, HIV SMRTcap enables a highly comprehensive characterization of HIV-1 reservoirs. Moreover, we demonstrate here that HIV SMRTcap performs robustly across the major global subtypes (HIV-1 subtype A, B, C, D and A/D recombinant viruses), and can use both cell- and tissue-derived inputs, including samples from antiretroviral therapy (ART) treated individuals with undetectable viral loads. Our results demonstrate that HIV SMRTcap serves as a comprehensive, robust method for unbiased HIV-1 reservoir characterization. Used alone, or in combination with single-cell based methods, HIV SMRTcap will enable novel exploration of viral reservoirs across subtypes and in tissue-specific compartments, providing critical information needed to inform HIV-1 cure.

## Author summary

Human Immunodeficiency Virus (HIV) remains a persistent global and lifelong infection because the virus integrates into the genome of infected cells and is then protected by host mechanisms. Characterization of these virus-harboring cells ("reservoir") has proven difficult due to a multitude of technical limitations, but remains critical as this viral reservoir reestablishes active infection upon therapy interruption. Understanding HIV integration patterns with regards to preferred genomic loci, prevalence of reservoir-containing cells within tissues, and profiling of internal deletions within the integrated viruses (proviruses) is pivotally important for vaccine and therapy design. In this work, we describe HIV Single Molecule Real Time Capture (HIV SMRTcap), an innovative pipeline that leverages long-read sequencing to simultaneously resolve proviral integration site(s) and integrity. This strategy enriches HIV-containing genomic DNA prior to single molecule sequencing and analyses using a custom informatic suite, ultimately capturing HIV at high sensitivity in contexts with and without antiretroviral therapy suppression, from primary tissue sources, and across HIV subtypes. We aim to describe HIV SMRTcap and its paired analysis tools in detail to enable wide adoption and encourage a pansubtype approach for equitable HIV research.

## Introduction

Despite four decades of advancements in therapy and care, Human Immunodeficiency Virus type 1 (HIV-1) remains a significant global public health challenge, affecting nearly 40 million people [1]. While the rate of new infections has declined from the epidemic's peak, > 1 million individuals are still newly infected each year [2,3]. Globally, there are multiple co-circulating subtypes that asymmetrically contribute to total burden and can differ by up to 30% at the nucleotide level [4]. Regardless of subtype, uncontrolled HIV-1 infection progressively depletes CD4 + T cells and dysregulates immune system function, and even treatment-suppressed individuals with HIV-1 experience higher rates of co-morbidities associated with chronic inflammation [5–7]. Even with effective antiretroviral therapy (ART) to limit progression of HIV-1, there remains no cure [8]. The primary obstacle to achieving a sterilizing or functional cure is the persistence of a persistent HIV-1 reservoir, arising from HIV-1 DNA integration into target host cell genomes, primarily activated T cells in the periphery [9,10]. Transition of these infected, activated cells to a resting state restricts proviral transcription and mediates the persistence of a quiescent, integrated proviral genome throughout the individual's lifespan [11,12]. Upon cellular activation and subsequent latency reversal, virus production reseeds the HIV-1 reservoir [13].

Different types of proviruses form as a result of HIV-1 integration. In >90% of cases, the integrated proviral genome contains internal deletions or other defects, rendering them unable to produce viable viral particles [14]. Although intact proviral genomes are required for the generation of infectious viral particles, defective proviruses can be transcribed and translated, triggering inflammation and immune responses in individuals with ART-suppressed HIV-1 [14,15]. Similarly, while resting CD4 + T cells are generally thought to be long-lived, it is the combination of antigen stimulation, homeostatic proliferation, and integration-site stimulated replication that triggers provirus-containing cell proliferation and maintains the overall reservoir load throughout the life of an infected individual [14,15]. Critically, HIV-1 cure-focused interventions target provirus-containing cells for removal. As such, viable cure strategies will need to be informed by high-resolution molecular and computational reservoir characterization methodologies. The cellular composition of the HIV-1 reservoir is also complex; in addition to circulating CD4 + T cells, HIV-1 can infect macrophages, dendritic cells, and microglia to form reservoirs within tissues including the liver, kidneys, lungs, lymphoid tissues, central nervous system, and gastrointestinal tract [16–18]. This broad tissue tropism contributes to the complexity of HIV-1 pathogenesis and reservoir characterization [19]. Whether rebound infection is initiated and/or driven in tissue-specific latent reservoirs remains unclear [15,20].

HIV-1 favors particular regions of the host genome for integration, including AT-rich regions and open chromatin in actively transcribed genes [21–23]. As a result, integration site proximity to genomic features such as introns or CpG islands may directly modify reactivation potential through disruption of native gene regulatory mechanisms [24–26]. Unfortunately, technical limitations associated with short-read, next-generation sequencing (NGS) have made it challenging to directly associate integration sites with proviral integrity and reactivation potential within the intact proportion of the HIV-1 reservoir [27,28]. While integration sites and proviral integrity can now be linked by advances in single-cell sequencing and single-virus barcoding of provirus-containing cells, these recent high-resolution analyses still rely on NGS and may be limited by genomic context and throughput [24,29–35]. This continued development of methods that link integration sites and proviral integrity status demonstrate the need for this level of resolution, which will likely prove pivotal to furthering our understanding regarding what is needed for reservoir-targeted cure strategies. However, there remain limitations to these methods, including limited specificity (as many strategies solely target HIV-1 subtype B), reliance on long terminal repeat (LTR)-specific or gene-specific amplification sites, and computational challenges identifying integration sites embedded within highly repetitive genomic regions that are not always sufficiently defined using NGS.

To collectively address these limitations, we have developed HIV Single Molecule Real Time Capture (HIV SMRTcap), a highly accurate, single-molecule HIV-1 reservoir sequencing and characterization pipeline that provides direct and simultaneous resolution of integration site and proviral integrity in an HIV-1 subtype-agnostic manner. With total genomic DNA (gDNA) as input, HIV SMRTcap performs robustly across both peripheral blood mononuclear cells (PBMC) and tissues,

with the specificity and sensitivity required to characterize HIV-1 reservoirs in both viremic and ART-suppressed samples. We believe that HIV SMRTcap will be a useful complimentary tool to provide global HIV-1 reservoir characterization with higher accuracy than comparable methods.

## Results

### Overview of the HIV SMRTcap method

As schematized in Fig 1 and described in the Materials and Methods, HIV SMRTcap aims to capture fragments of high-molecular-weight (HMW) gDNA carrying integrated proviral HIV-1 content using a pan-subtype oligonucleotide pool and then subject these to single-molecule sequencing and annotation for downstream analyses. Briefly, HMW gDNA is mechanically sheared prior to barcoding and non-HIV-specific amplification. The resulting material is size-selected to enrich for fragments larger than 8 kilobases (kb) and then hybridized to our custom cross-subtype HIV-1-specific probe set. Captured DNA fragments containing HIV-1 proviral content are enriched using streptavidin-conjugated magnetic beads, and then non-specifically amplified to increase copy number for sequencing. DNA libraries are prepared and processed using single-molecule, real-time sequencing, generating long reads with an average accuracy of 99.98%. These reads are then iteratively mapped to a library of multiple full-length HIV-1 subtype reference sequences to remove non-specific data (Fig 1B).

Reads containing HIV-1 content are then mapped to the human reference genome (hg38) for integration site identification (Fig 1C). Mapping at this stage also provides integration site information regarding genomic feature, including ENCODE elements, repetitive elements, and CpG islands. Unique shearing sites provided by mechanical fragmentation act as "endogenous barcodes", allowing for the discrimination and filtering of PCR duplicates [36]. Subsequently, all collapsed PCR reads with the same integration site are labeled as a single clone, allowing for quantification of clonal expansion, a key driver of HIV persistence (Fig 1C). Proviral integrity of each unique viral clone is performed on a per-viral-gene basis, identifying the presence/absence/partial content of each viral gene to assign to one of five distinct integrity categories. These categories include: (i) "Intact" – single molecule reads that contain both human genomic flanks with HIV-1 proviral content contained within that has > 75% presence of all 11 viral genes; (ii) "Putatively Intact" – reads that contain only one (5' or 3') human genomic flank due to random mechanical shearing but retain significant proviral content such that we presume proviral intactness based on completeness of all present genes; (iii) "Internal deletions" – reads that are either dual- or single-flanked by the host genome and contain sufficient proviral content to observe internal viral gene deletions within sequenced DNA; (iv) "Truncated" – reads with flanking human genomic content that contains one or more viral genes present and appears to "run into" the human genome; and (v) "Indeterminate" – reads that contain HIV-1 and human flanking content but were sheared such that there is insufficient viral content to assess proviral integrity (schematized in Fig 1C). Indeterminate reads are used for integration site annotation only and not used for further proviral characterization. Each viral gene is assigned a score to indicate their complete deletion ("0"), partial deletion (0–75% mapped, "1"), high to complete intactness (>75% mapped, "2"), or absence due to shearing ("-") in the downstream HIV SMRTcap data reports. All data per single molecule sequenced, including integration site and genomic context, proviral integrity, extent of clonal expansion, and the nucleotide-resolved sequence of both the human flanks and proviral insert are reported in a master summary sheet (.xlsx) and accompanying FASTA files (S1 Table). Samples analyzed in this demonstrative study are summarized in Table 1, and sequencing metrics associated with their HIV SMRTcap data are summarized in Table 2.

### Initial validation of HIV SMRTcap

We initially set out to demonstrate the performance of HIV SMRTcap by characterizing HIV-1 proviral integration in the 8e5 cell line, which persists with an HIV-1 proviral integration in chromosome 13 (chr13; [37] Fig 2A). As expected, our analysis identified that 99% of clones detected used this primary proviral integration site. The known chr13 integration was

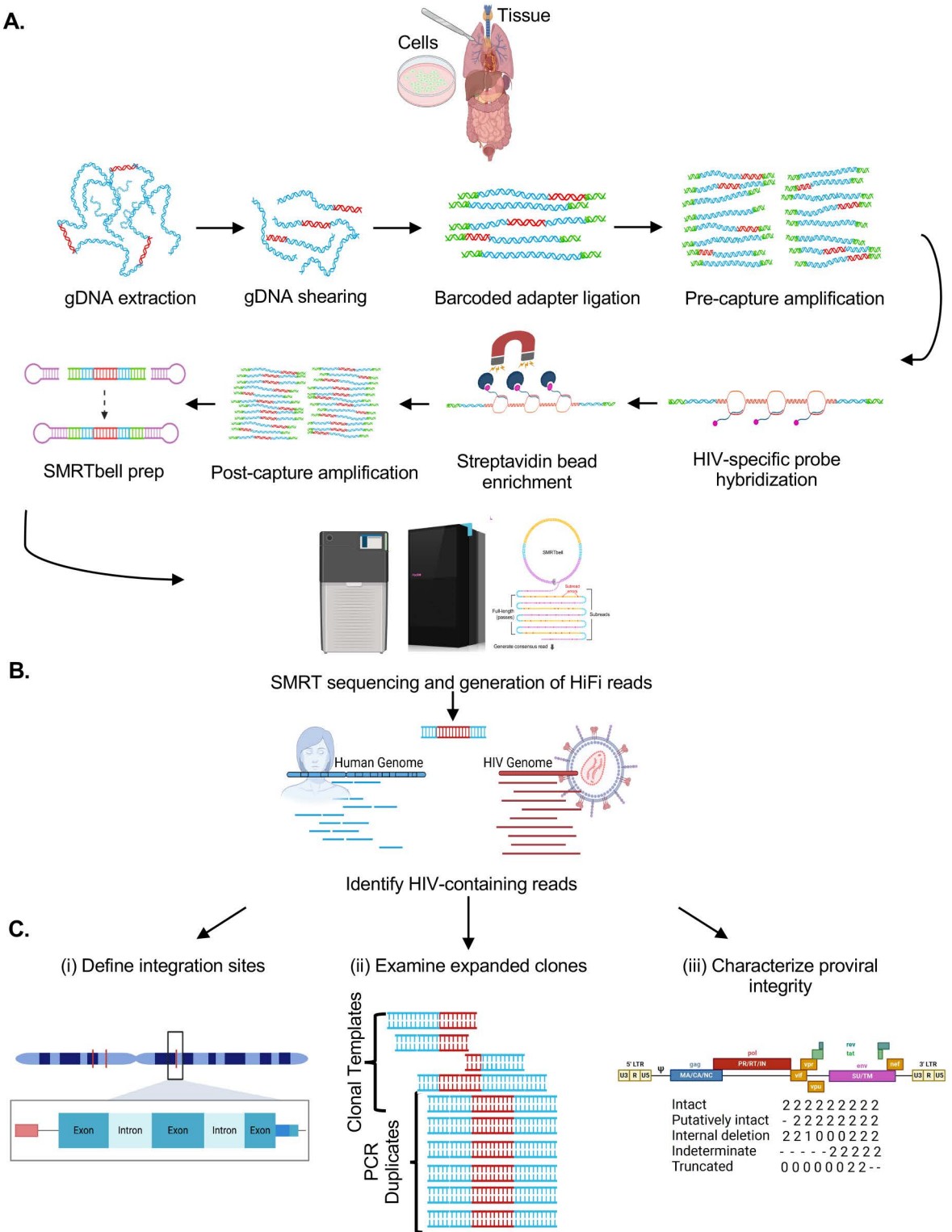

**Fig 1. Schematic of HIV SMRTcap pipeline. (A)** High molecular weight genomic DNA (gDNA) is extracted from either a cell- or tissue-based source. Integrated HIV-1 provirus is represented in red, with non-specific host gDNA represented in blue. Barcoded universal adapters are green and are used for non-HIV-1-specific amplification of total gDNA pre- and post-oligo-based capture. Once bound to HIV-1-specific probes, enriched with

streptavidin-based beads, and amplified, the HIV SMRTcap products are then adapted into a SMRTbell sequencing template via the ligation of hairpin adapters, represented in pink. Sequencing was performed on either the Sequel IIe or Revio platform (see Table 2), and high-fidelity intramolecular consensus ("HiFi") reads were generated on instrument. **(B)** Primary analysis included identification, selection, and binning of reads that contain at least 500 bp of continuous HIV-1 genomic content for further examination. **(C)** Characterization of HIV-1-containing reads enriched by the SMRTcap method, includes (i) identification and classification of integration sites; (ii) definition of expanded clones and filtration of PCR duplicates; and (iii) assessment of proviral integrity using a gene-by-gene mapping approach. Created in BioRender. Sadri, **G.** (2025) https://BioRender.com/at320g3.

**Table 1. Samples used for HIV SMRTcap validation and associated subject metadata.**

| Sample ID | Sex (Donor) | Race | Age | Sample Type | HIV Status/ Subtype | ART Status | Last IPDA | Last Viral Load |
|---|---|---|---|---|---|---|---|---|
| **TISSUES** | | | | | | | | |
| NNTC: Basal Ganglia | Male | Hispanic | 42 | Brain tissue - Basal Ganglia | HIV B+ | Remote administration, none at death | N/A | 162642 |
| NNTC: Heart | Male | Black | 65 | Heart Tissue | HIV B+ | ART Suppressed | N/A | 30 |
| NNTC: Basal Ganglia | Male | Hispanic | 21 | Brain tissue - Frontal Lobe | HIV Negative | N/A | N/A | N/A |
| Mouse_P1 | Male | Hispanic White | 32 | Xenograft Mouse Spleen (NSG) Splenic CD4+T Cell | HIV B+ (Donor)/ HIV JR-CSF (Mouse) | ART Suppressed (Donor), ART Naïve (mouse) | 2.23E+05 (mouse spleen) | Donor: <20/ Mouse: 3.53E+07 copies/mL |
| HOPE P4 | Male | Black | 59 | PBMC | HIV B+ | ART Suppressed | 594.21/1M | <50 (UD) |
| Donor 13 (12)* | Female | Black | 47 | PBMC | HIV A1D+ | ART Suppressed | N/A | <50 (UD) |
| Donor 13 (14)* | Female | Black | 49 | PBMC | HIV A1D+ | ART Suppressed | N/A | 85 |
| Donor 13 (15)* | Female | Black | 50 | PBMC | HIV A1D+ | ART Suppressed | N/A | <50 (UD) |
| Donor 13 (17)* | Female | Black | 52 | PBMC | HIV A1D+ | ART Suppressed | N/A | <50 (UD) |
| Donor 17 (9)* | Female | Black | 41 | PBMC | HIV A1+ | ART Suppressed | N/A | <50 (UD) |
| Donor 17 (11)* | Female | Black | 43 | PBMC | HIV A1+ | ART Suppressed | 2640.5/200367 | <50 (UD) |
| Donor 17 (12)* | Female | Black | 44 | PBMC | HIV A1+ | ART Suppressed | N/A | <50 (UD) |
| Donor 17 (14)* | Female | Black | 46 | PBMC | HIV A1+ | ART Suppressed | N/A | <50 (UD) |
| **CELL LINES** | | | | | | | | |
| 92UG_029 | N/A | N/A | N/A | Cell line | HIV A+ | N/A | N/A | N/A |
| 89BZ_167 | N/A | N/A | N/A | Cell line | HIV B+ | N/A | N/A | N/A |
| 93MW_965 | N/A | N/A | N/A | Cell line | HIV C+ | N/A | N/A | N/A |
| 93UG_065 | N/A | N/A | N/A | Cell line | HIV D+ | N/A | N/A | N/A |
| 8e5 | N/A | N/A | N/A | Cell line | HIV B+ | N/A | N/A | N/A |
| SupT1_R5 | N/A | N/A | N/A | Cell line | HIV Negative | N/A | N/A | N/A |
| CEM | N/A | N/A | N/A | Cell line | HIV Negative | N/A | N/A | N/A |

also observed in multiple passages of cultured 8e5 cells at similarly high rates (99.62%±0.13%; Fig 2B). While this cell line is often considered relatively clonal because the canonical HIV provirus integrated in chr13 contains a point mutation in *pol* that renders it most defective, low-level and function-restoring reversions have been described upon serial passage [38,39]. HIV SMRTcap did detect these low level integration events in both non-repetitive and repetitive regions across the genome (Fig 2B). Additional integration sites were also observed at passage 2, which we assume reflects prior low-level

**Table 2. Sequencing platforms used and read metrics for all samples presented.**

| Sample ID | HIV-SMRTcap total CCS reads | HIV-SMRTcap total HIV reads | HIV-SMRTcap unique HIV reads | Multiplexing | Sequencing System |
|---|---|---|---|---|---|
| **TISSUE** | | | | | |
| 010011_BR | 900173 | 4313 | 2001 | 4-plex | Sequel IIe |
| MHBB676_HT | 949009 | 49464 | 16132 | 4-plex | Sequel IIe |
| MHBB588_BR | 1158464 | 0 | 0 | 4-plex | Sequel IIe |
| BJ_Mouse_P1 | 825158 | 16515 | 6829 | 4-plex | Sequel IIe |
| 160_2 Uganda | 11508884 | 293 | 166 | Single | Revio |
| 160_4 Uganda | 723405 | 533 | 166 | 4-plex | Sequel IIe |
| 160_5 Uganda | 907561 | 311 | 141 | 4-plex | Sequel IIe |
| 160_7 Uganda | 166178 | 425 | 212 | Single | Revio |
| 171_2 Uganda | 1032783 | 531 | 199 | 4-plex | Sequel IIe |
| 171_4 Uganda | 682341 | 120 | 45 | 4-plex | Sequel IIe |
| 171_5 Uganda | 880430 | 158 | 64 | 4-plex | Sequel IIe |
| 171_7 Uganda | 178770 | 120 | 36 | Single | Revio |
| **CELL LINES** | | | | | |
| 92UG_029 | 487219 | 124581 | 56602 | 2-plex | Sequel IIe |
| 89BZ_167 | 1169298 | 18226 | 6895 | 3-plex | Sequel IIe |
| 93MW_965 | 1300346 | 53647 | 16988 | 4-plex | Sequel IIe |
| 93UG_065 | 596352 | 148810 | 43041 | 4-plex | Sequel IIe |
| 8e5_P2 | 719848 | 71228 | 9502 | 4-plex | Sequel IIe |
| 8e5_P44 | 726508 | 45214 | 30026 | 4-plex | Sequel IIe |
| 8e5_P86 | 717945 | 31831 | 16212 | 4-plex | Sequel IIe |
| 8e5_P112 | 579282 | 25114 | 11235 | 4-plex | Sequel IIe |
| LoD 1-8e5/CEM = 1:100* | 2126934 | 311 | – | 4-plex | Revio/Sequel IIe |
| LoD 2-8e5/CEM = 1:1000* | 1442621 | 38 | – | 4-plex | Revio/Sequel IIe |
| LoD 3-8e5/CEM = 1:10000* | 1783620 | 5 | – | 4-plex | Revio/Sequel IIe |
| LoD 4-8e5/CEM = 1:100000* | 1694460 | 11 | – | 4-plex | Revio/Sequel IIe |
| SupT1_R5 | 243347 | 0 | 0 | 3-plex | Sequel IIe |
| CEM | 555232 | 0 | 0 | 4-plex | Sequel IIe |

*= average (n = 4).

passage history necessary to maintain cell line stocks. The overall number of HIV-1 proviruses captured from 8e5 cells decreased after extensive passage, despite identical material input (Table 2). Although counterintuitive, this result agrees with prior studies, which have shown that extended 8e5 passage results in a loss of HIV proviral content [38].

To orthogonally validate the sensitivity of HIV SMRTcap, we compared the identified integration sites in the 8e5 cell passages to those identified by ligation-mediated polymerase chain reaction (LM-PCR) (Fig 2C; [28]). In this LM-PCR approach, asymmetric linkers were directly ligated to sheared gDNA, followed by two rounds of LTR-linker-based PCR amplification. These libraries were then sequenced by 150 bp paired-end Illumina sequencing for downstream bioinformatic analyses. Overall, many integration sites, including the dominant chr13 provirus, were identified by both HIV SMRTcap and LM-PCR. Not surprisingly, given the different amplification procedures and sequencing methodologies, low-level minor integrations did not completely agree between methods, as each approach analyzed distinct gDNA templates from the same preparation. Interestingly, > 90% of reads from HIV SMRTcap mapped to the known chr13 integration site, whereas only ~25% of LM-PCR reads identified this region. To ensure that the additional integration events we observed

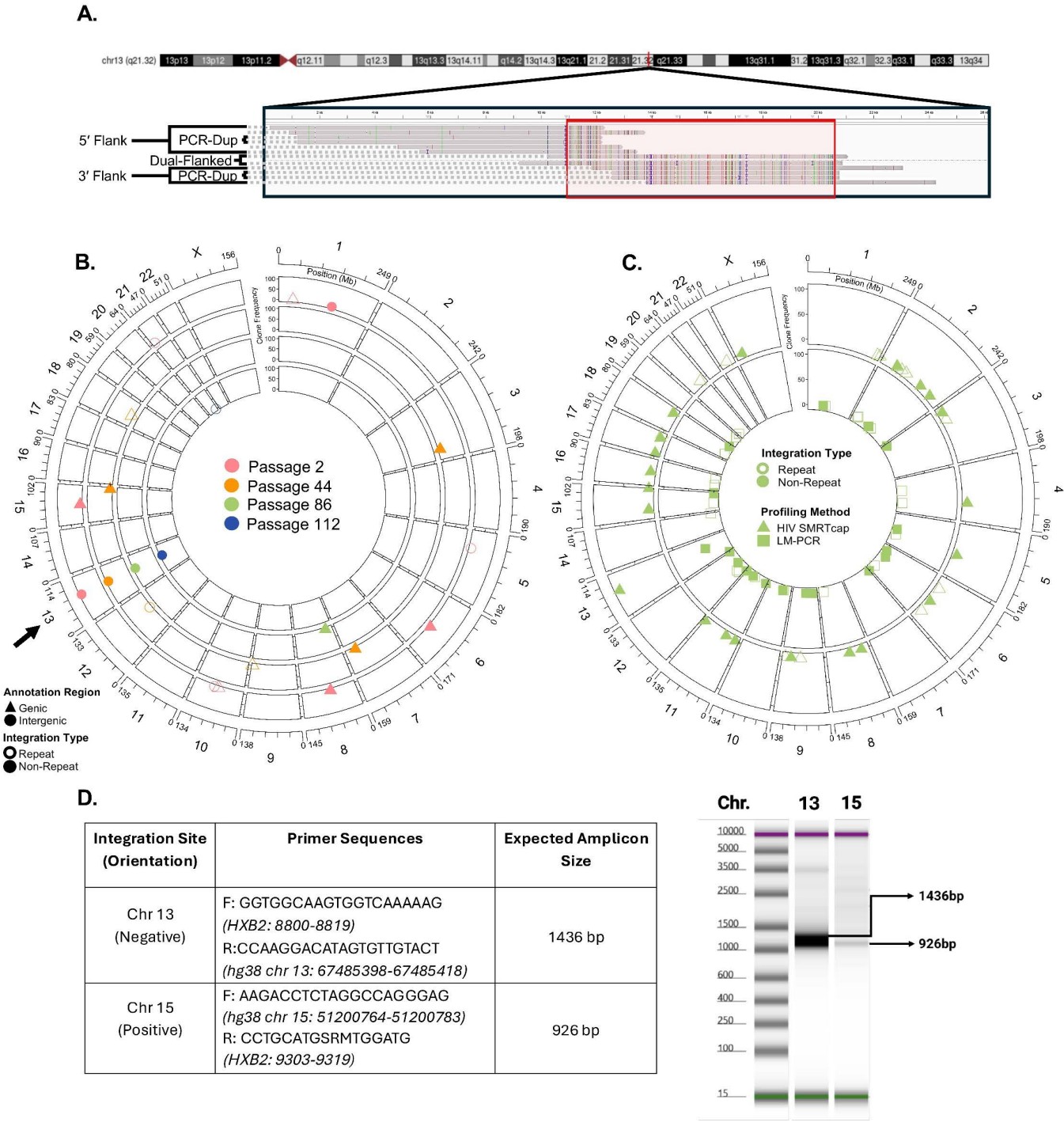

**Fig 2. Validation of HIV SMRTcap. (A)** Multi-sequence alignment of selected HIV SMRTcap reads generated from 8e5 cells carrying a known integration in chromosome 13 (chr13). The long-read, single molecule reads illustrate examples of sheared-end usage for identifying PCR duplicates, and visualization of the three HIV SMRTcap data types: (i) 5'-flanked; (ii) 3'-flanked; and, (iii) dual-flanked HIV-1 proviruses, which share the same integration site and therefore belong to the same clone. Colored bars indicate mismatches versus consensus hg38 and HIV-1 strain HXB2 sequences. Proviral sequence is denoted by the genomic content contained within the red box. **(B)** Evaluation of 8e5 integration sites identified by HIV SMRTcap across four serial passages: passage 2, 44, 86 and 112 are represented by pink, orange, green, and blue symbols and moving from outer to inner chords, respectively. Open symbols indicate integration within repetitive genomic regions, and filled shapes denote integration within non-repetitive regions. Triangles

identify integrations within genes, and circles denote integrations in intergenic regions. The Y-axis represents the number of unique reads sharing the same integration site (clone size). The black arrow identifies the expected main integration event on chr13. **(C)** Orthogonal validation comparing identification of integration sites by HIV SMRTcap and LM-PCR using the same 8e5 cell-derived gDNA input, specifically from passage 86. Triangles on the outer chord indicate integration sites identified by HIV SMRTcap, while the inner chord displays integration sites detected by LM-PCR in squares. Clone size is represented on the Y-axis, and open/closed symbols again identify integration sites in repeat or non-repetitive genomic regions, respectively. **(D)** Molecular validation of a selected novel integration site on chromosome 15 on a TapeStation gel using PCR amplification with primers based on HIV SMRTcap data. The known chr13 integration site was included as a positive control.

with HIV SMRTcap, but not LM-PCR, were present in our 8e5 cell line (passage 86) and not an analysis artifact, we designed targeted amplification primers to the host gDNA, and HIV-1 *gag* based on our HIV SMRTcap data at chr13 (positive control), and chr15 (HIV SMRTcap-only integration site). Fig 2D shows that these primers successfully targeted and amplified this example of a novel integration to a much lower endpoint amount than the positive control.

### HIV SMRTcap enriches proviral reservoirs regardless of subtype

A critical motivation in our development of HIV SMRTcap was to create a pipeline that performed robustly across HIV-1 subtypes. Current HIV-1 reservoir characterization methods often must be completely redesigned, optimized, and validated when extending capacity to non-B HIV-1 subtypes [40]. We developed the HIV SMRTcap oligo capture pool to represent multiple genomes from all known subtypes such that it can be used to characterize reservoirs from samples with unknown HIV-1 subtype etiology and viral recombinants. To demonstrate pan-subtype efficacy, we assessed the ability of HIV SMRTcap to characterize integrated provirus from *in vitro* infections of SupT1-R5 or SupT1-R4 cells across the four major global HIV-1 subtypes (A, B, C, and D) (Fig 3; [41–43]). Fig 3A demonstrates that the pan-subtype design of HIV SMRTcap efficiently targeted and resolved HIV-1-containing integration events across HIV-1 subtypes A, B, C, and D. As expected, proviral integrations were observed primarily in intronic regions, with a small proportion of integrations in intergenic regions. gDNA from uninfected SupT1 cells were used as negative controls to assess off-target effects; captured sequences from these samples were entirely off-target, enriched mostly simple repeats, and did not demonstrate preferential enrichment of host endogenous retroviruses, such as HERV-K (Table 2 and S1 Fig).

Given the enhanced resolution of integration site identification in genomic repeats provided by the longer flanking regions captured by HIV SMRTcap, we also assessed integration frequencies into common genomic repeats across subtypes (Fig 3B). All subtypes demonstrated integrations across a variety of repetitive genomic elements, as has been described [44]. There did not appear to be subtype-specific biases or patterns for these repeat-based integrations. As a function of the total integrations, repeat-based events represented approximately ~49% (range: 48–50%) of integrations; likewise, genic integrations represented ~81.5% (range: 79.4-84.6%) of all integrations. These data highlight the importance of including repeat-resolved integration site mapping to comprehensively characterize primary HIV-1 reservoir samples, as it enables mapping of proviral DNA to more of the genome.

Nucleotide-resolved resolution of the HIV-1 proviral genome enabled us to examine integrity in a gene segment-by-gene segment manner and to quantify the total content per gene that was present across the proviruses recovered in our data (Fig 3C). To note, due to the randomness of mechanical gDNA shearing, there were always reads that were judged "indeterminate" due to missing content when shearing occurred up- or down-stream from that gene. If gene content was identified, it was then evaluated for the total presence at three levels, > 75% of the gene was present, between 0% and 75% of the mapped content was present, or gene was completely deleted. The relative pattern of proviral internal deletions did appear different with 93-RW (HIV-1 subtype A) versus subtypes B, C and D. Notably, 93-RW/subtype A showed more extensive small deletions of HIV-1 accessory genes *vif, vpr, and vpu* alongside a notable difference in structural gene *gag,* than the other viruses. To note, computational analysis for these samples used the corresponding NCBI reference for each unique virus infection; therefore, small deletions observed are not an artifact of mapping to HXB2.

PLOS Pathogens

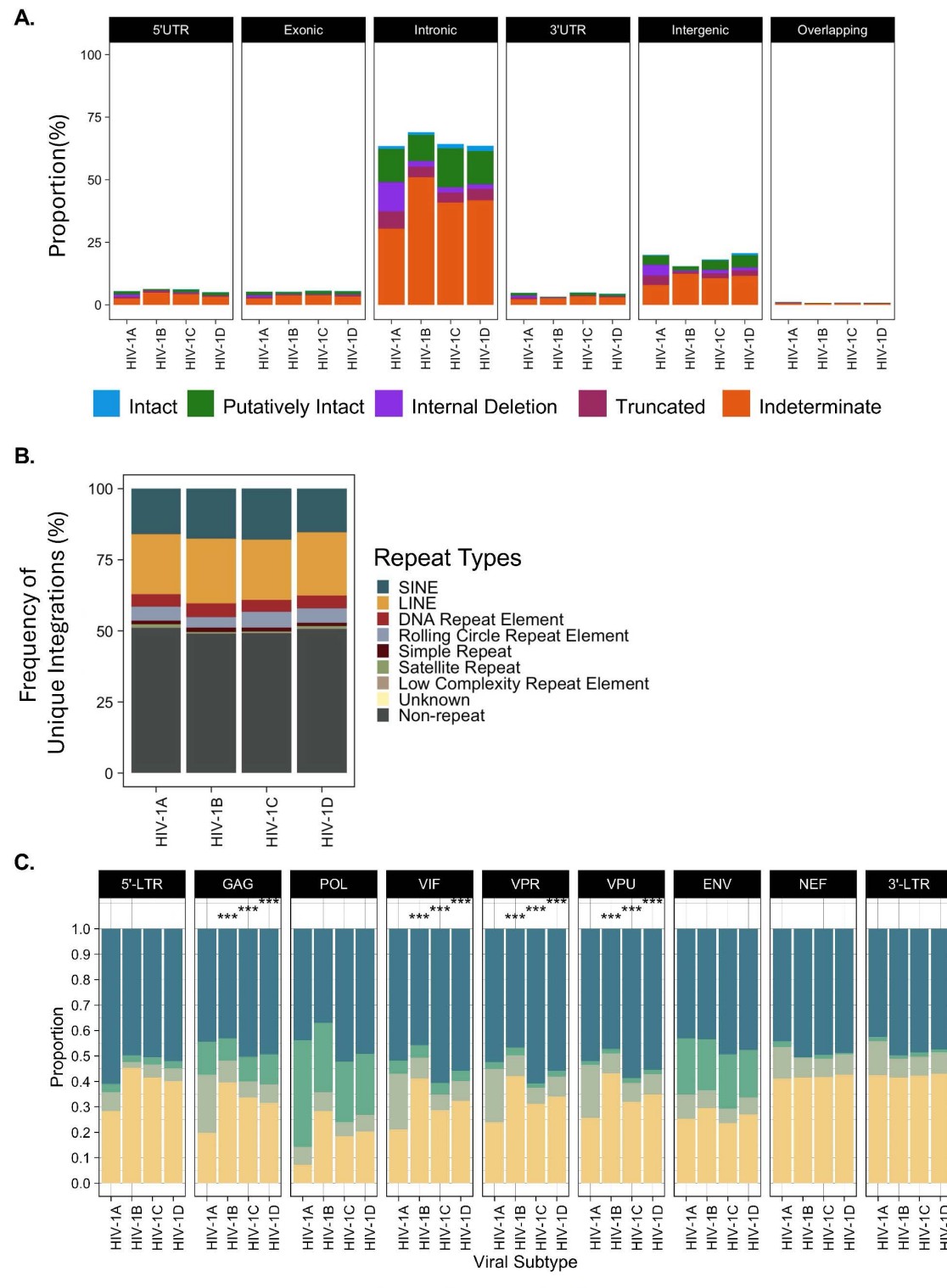

**Fig 3. HIV SMRTcap Performance across four major HIV-1 subtypes. (A)** Stacked bar graphs demonstrating integration sites identified with HIV SMRTcap from cells infected with each of four major HIV-1 subtypes. Stacked bars indicate integrations sites per genomic region (black headers) with

the relative height of the bar representing the proportion of reads classified as intact (blue), putatively intact (green), internal deletions (purple), or truncated (maroon). Indeterminate reads that do not inform on proviral integrity due to random shearing are orange. **(B)** Relative frequency of unique HIV-1 integration events occurring in non-repeat and varied genomic repeat elements per indicated subtype. **(C)** Detailed gene segment-by-segment evaluation of proviral integrity across HIV-1 subtypes. Each segment was classified by the extent of detected gene content when aligned to the matched viral reference. Dark blue indicates >75% expected gene content present; green indicates between 0% and 75% gene content detected; taupe indicates the gene is missing entirely, but flanking content is present indicating deletion; yellow specifies the relative frequency of reads where detectable gene content was lost due to shearing. *** indicates statistically significance (p < 0.001) in two-way comparisons between accessory gene deletions found in subtype A vs subtype B, subtype A vs subtype C, or subtype A vs subtype D, using a chi-squared tests with Bonferroni correction for multiple testing.

Further, we observed a consistent proportion of viruses across subtypes that had significantly reduced or absent 3' LTRs. Although a relatively small fraction of total recovered 3' LTR sequences, it is notable considering the role expected for 3' LTR sequences in multiple steps of the viral lifecycle, including integration, transcription of latency-driving ncRNA, and polyadenylation of pre-mRNA [45,46]. Unrelated studies have also described the recovery of solo-LTR sequences from patient samples [47]. These results highlight that HIV SMRTcap enriches proviral genomes regardless of LTR-sequence or content variations. Overall, we observed that the core of the viral genome, including structural and non-structural genes, is often maintained with most of their content and as intact. Our data suggest that structural differences could exist across subtypes but would require extensive validation from extensively expanded primary samples.

### HIV SMRTcap performance in primary tissue samples

It is estimated that only 2% of the body's T cells are present in the blood at any time, with the majority circulating in and out of tissues and lymphatic structures surveying for pathogenic insult [48]. Tissue-localized HIV-1-reservoirs include T cells that can undergo activation, viral production, and tissue-specific evolution or compartmentalization in the absence of ART; in some organ systems such as the central nervous system, there is additional evidence of a myeloid cell compartment that contributes to the reservoir [16,49]. Comprehensive exploration of tissue-localized HIV-1 reservoirs has been hampered by sample availability and methodological innovations allowing for reservoir resolution from bulk tissue without the costly steps of single-cell suspension and limiting dilution. We examined the ability of HIV SMRTcap to resolve tissue-based reservoirs across tissues sourced from post-mortem HIV-1 subtype B infections collected by the Manhattan HIV Brain Bank (member of the National Tissue Consortium), and from humanized NSG mice infected with the HIV-1 subtype B isolate, JR-CSF (€).

We first examined the HIV-1 reservoir composition from the basal ganglia of a viremic tissue donor with HIV-1 who had a terminal plasma viral load (VL) of 162,642 IU/mL, having been off ART for 8 months prior to death. As expected from an individual with a history of ART treatment, most viral sequences present contained internal deletions (Fig 4A). Several expanded clones were observed at intronic and intergenic sites. The extensive low-level integration across the genome suggests high VL at the time of death (Fig 4A).

We also examined the reservoir composition in the heart tissue of a distinct HIV-1-infected individual who clinically presented with congestive heart failure (Fig 4B). We observed extensive viral integration in the heart, a tissue not often considered an HIV-1 sanctuary, particularly since plasma VLs two weeks before death were low (30 copies/mL) and ART was discontinued only 5 days before death in this donor. We then examined the post-mortem cardiac pathology for an explanation of this finding, which revealed dilated cardiomyopathy with a mixed inflammatory pericarditis, epicarditis, and myocarditis. Therefore, we speculate that the inflammatory changes found in the heart at the time of death contributed to homing of systemic T cells, many of which carried HIV-1 proviruses, and that this inflammation accounted for the reservoir load we observed with HIV SMRTcap.

Lastly, we examined the HIV-1 reservoir in enriched human CD4 + T cells isolated from a participant-derived xenograft (PDX) mouse model (Fig 4C). A NOD scid gamma (NSG) mouse was reconstituted with 5 million total memory CD4 + T cells from a human viremic controller donor (viral load history documented in S2 Table), followed five weeks later by high

PLOS Pathogens

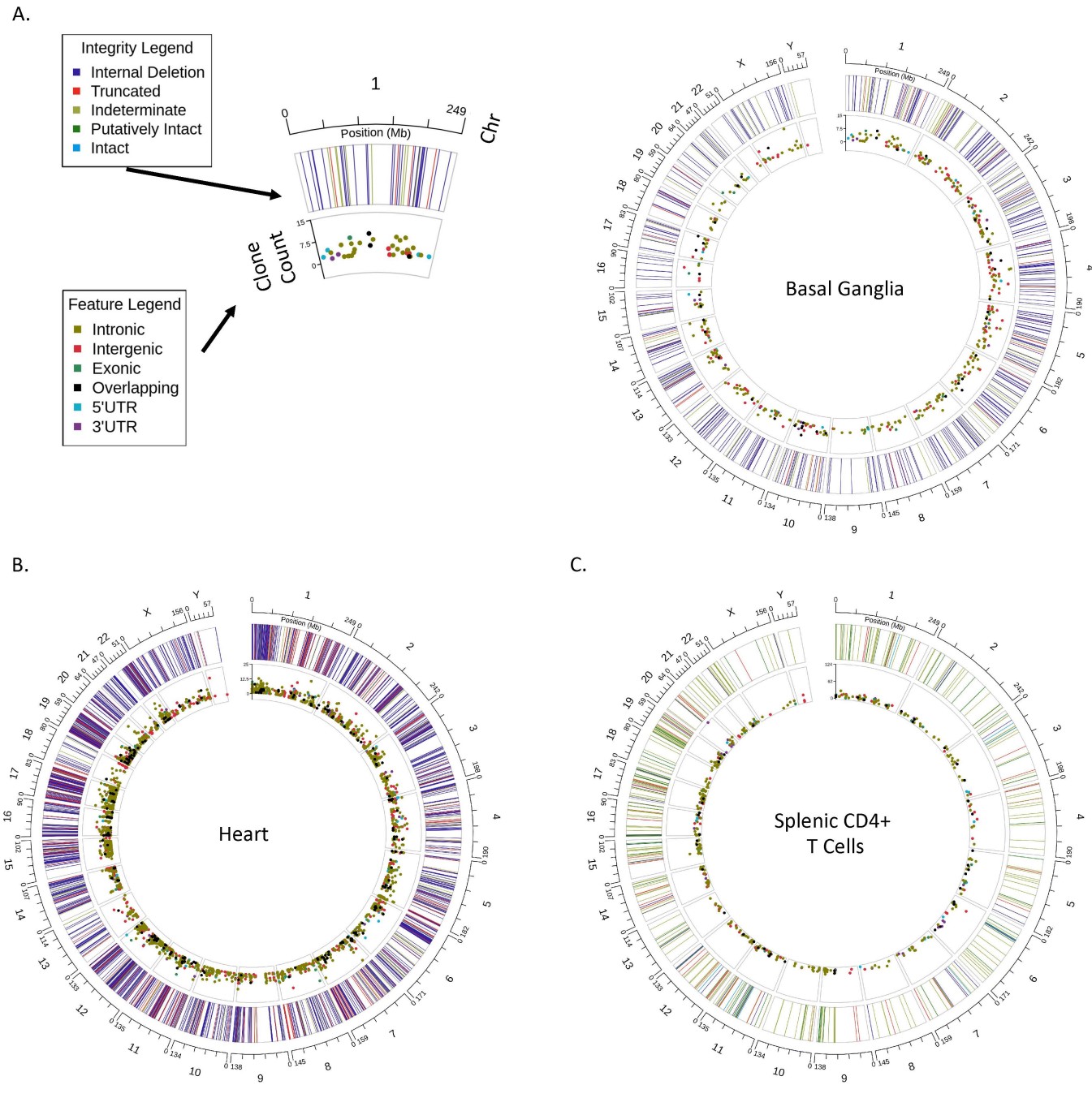

**Fig 4. HIV SMRTcap performance across tissues.** Circos plots represent reservoir profiles from diverse tissues captured by HIV SMRTcap using bulk gDNA as input: **(A)** basal ganglia, **(B)** heart, and **(C)** splenic CD4+T cells isolated from a humanized mouse model. The Circos plot format allows visualization of the multiple features of HIV SMRTcap data concurrently: (i) the outermost hashed ring represents human chromosomes; (ii) each line within the outer ring corresponds to unique HIV-1 integration sites at their respective chromosomal locations, with line color indicating the proviral genome integrity at the corresponding integration site (see Integrity Legend); and (iii) the Y-axis of the inner ring represents the size of clonal lineages associated with each integration site, with the dot color denoting integration site genomic feature(s) (refer to Feature Legend).

titer infection with the HIV-1 subtype B strain JR-CSF. Eight weeks following HIV-1 infection, tissues were harvested, and total spleen cells were isolated for reservoir characterization. As no CD8 + T cells were used for reconstitution and the mouse was ART naive, the viral infection was allowed to propagate, which is reflected by the extensive and variable reservoir observed (Fig 4C). HIV SMRTcap efficiently captured intact/putatively intact proviruses and identified dominant expanded clones in chr6, chr15, and chr20. When examined phylogenetically, two statistically distinct viral outgroups were observed alongside the major infecting species, JR-CSF (S2 Fig). Although we did not sequence the virus of the viremic controller used for reconstitution, it is possible, given the genetic diversity of these outgroups, that one or both represent the original infecting viral swarm from the donor. All three groups were classified as subtype B. Together these data demonstrate robust performance and feasibility of HIV SMRTcap using bulk HMW gDNA isolated from total tissue, tissue-localized sorted cell populations, and in exploring novel questions in the context of pre-clinical models of HIV-1 infection.

## HIV SMRTcap limit of detection

A major challenge for any HIV-1 reservoir characterization strategy is the comprehensive resolution of HIV-1 proviral DNA in ART-suppressed individuals, where the reservoir size is known to be particularly small [50]. To evaluate the ability of HIV SMRTcap to enrich and characterize low frequency proviral templates, we evaluated potential assay limits of detection leveraging serial 10-fold dilutions of HIV-1 provirus-containing 8e5 cells into CEM cells, an immortalized, HIV-negative T cell line (Fig 5).

We first modeled our limit of detection using conservative calculations of template input followed by the known subsampling that occurs at each step of the HIV SMRTcap protocol (Fig 5A). We performed these calculations assuming a milieu of sheared ~10kb gDNA fragments, as this is the environment in which the oligo-based enrichment must perform with high sensitivity. For example, an initial input of 10,000 copies of HIV-1 in a total input of ~$6.4 \times 10^{12}$ sheared gDNA fragments simulates a spike-in experiment equivalent to ~1 viral genome per $10^8$ gDNA templates. Informed by empirical quality control measurements during assay development, we extrapolated the subsampling of input material at each stage of the molecular preparation to estimate the final HIV copy number in the sequencing data. Although template loss does occur, our calculations indicated that recovery should remain sufficient to detect HIV-1 at extremely low frequencies (>$1 \times 10^{-10}$ sheared gDNA fragments) (Fig 5B).

HIV SMRTcap was performed in four replicates per 8e5:CEM dilution set to generate a rarefaction curve and assess actual results as compared to our predictions. These replicates included sequencing replicates of the same library preparation (n1, n2), a technical replicate from the same extracted gDNA (n4), and cell batch replicates from a unique 8e5:CEM dilution set to assess rigor and reproducibility (n3) (Fig 5B-5D). Aligned with our predicted detection limit of $1 \times 10^{-10}$, we empirically detected HIV-1 proviral DNA in 3 out of 4 replicates at this dilution, as well as in 3 of 4 replicates of a further ten-fold dilution set (~$1 \times 10^{-11}$). HIV-1 was undetectable in one replicate (n3) at both lower dilutions, likely due to reduced sequencing depth (~10-fold). Although we observed slightly higher rates of HIV-1 proviral detection at the lowest (~$1 \times 10^{-11}$) dilution compared to that at ~$1 \times 10^{-10}$, we believe these results indicate the impact of sampling bias, sequencing depth differences, and the exacerbation of detection limit variation at such low template frequencies.

## HIV SMRTcap characterization of ART-suppressed primary samples

We obtained two sets of samples to evaluate the ability of HIV SMRTcap to characterize the reservoir in ART-suppressed samples. These included ART-suppressed PBMC samples collected from an individual living with HIV-1 subtype B ("P4") who had undergone kidney transplant, which were collected at the time of transplant as part of the HOPE multicenter study, and samples collected longitudinally at four timepoints from two Ugandans living with HIV-1, one infected with HIV-1 subtype A1 and another with an HIV-1 subtype A1/D recombinant virus ("Donor 17" and "Donor 13", respectively) who were closely followed as part of the Rakai Latency Cohort in Rakai district, Uganda [40,51–55]. All individuals had

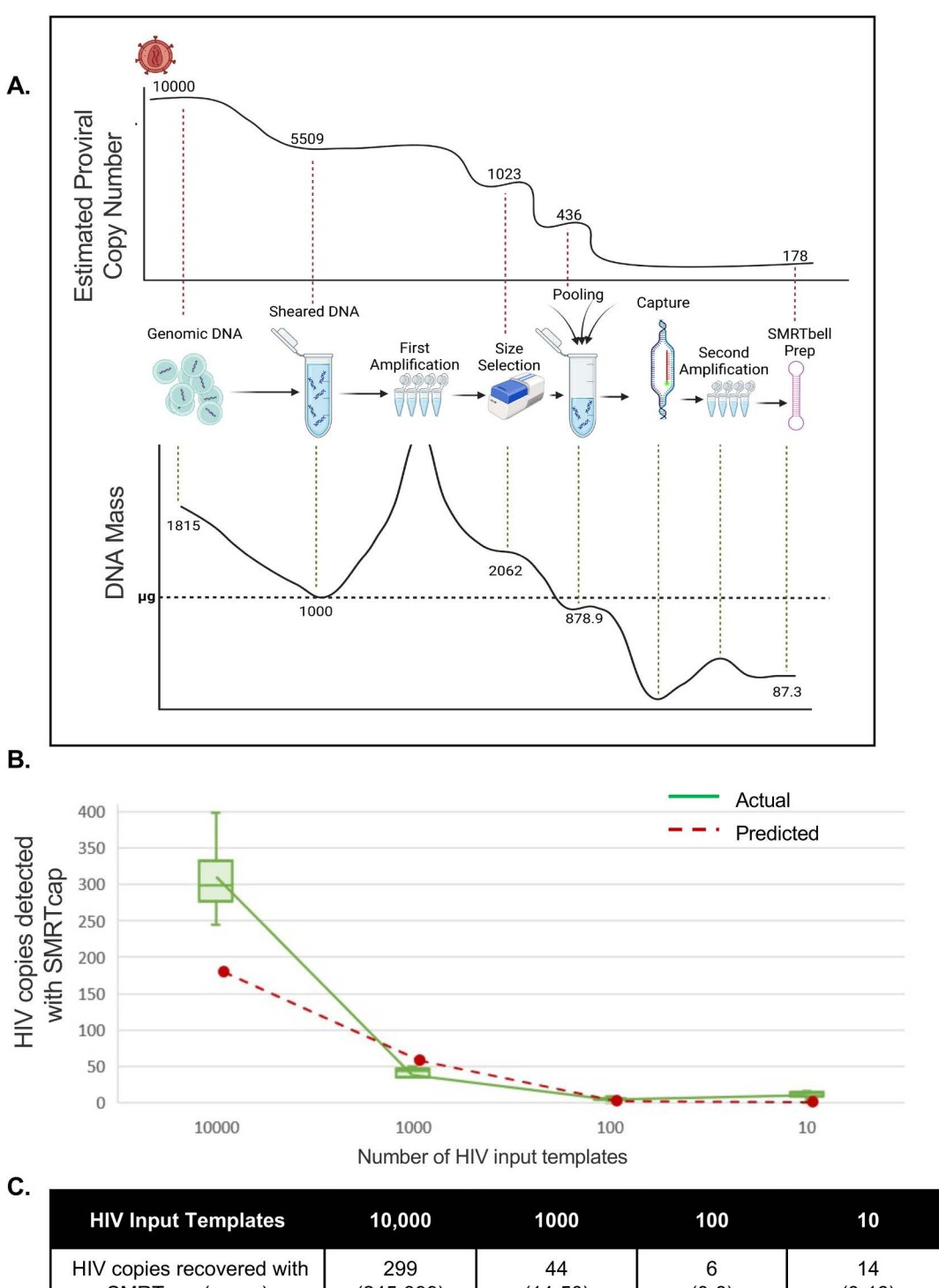

C.

| HIV Input Templates | 10,000 | 1000 | 100 | 10 |
|---|---|---|---|---|
| HIV copies recovered with SMRTcap (range) | 299 (245-399) | 44 (14-50) | 6 (0-8) | 14 (0-16) |
| Predicted Recovery | 178 | 57 | 2 | 0 |

**Fig 5. HIV SMRTcap limit of detection and subsampling. (A)** Subsampling of input gDNA material that occurs during the HIV SMRTcap process was modeled in the top curve, using the largest number of HIV-1 templates (10,000 HIV-1-containing cells) used as input into the limit of detection assay as an example. Stepwise subsampling is indicated by copy number decreases at each step of the protocol, concluding with the predicted final copy number

used as input into sequencing is presented. The red horizontal dashed line represents 1000 ng DNA. **(B)** Empirical HIV SMRTcap data was collected to experimentally establish assay limits of detection. An initial spike-in of 10,000 8e5 cells was made into HIV-negative CEM for a total of $10^6$ cells (1 HIV genome per $10^8$ of sheared 10kb gDNA fragments). Four 10-fold serial dilutions were processed with HIV SMRTcap to generate the rarefaction curve. Green indicates the number of HIV-1 templates detected by HIV SMRTcap at each dilution point; each dilution point was assessed with four replicates to assess technical and biological rigor and reproducibility. The red line represents the predicted HIV-1 copy numbers calculated for each dilution according to the subsampling schema established in **(A)**. **(C)** Empirically resolved HIV templates for each dilution and replicate compared to the predicted values. **(D)** Total number of CCS reads generated per replicate run allowing for evaluation of the impact of sequencing depth on HIV SMRTcap efficiency. Fig 5A Top: Created in BioRender. Sadri, **G.** (2025) https://BioRender.com/89hn9lt. Fig 5A Bottom: Created in BioRender. Sadri, **G.** (2025) https://BioRender.com/wtnujt3.

undetectable VL at the timepoints evaluated with HIV SMRTcap and had been undetectable for several years leading up to this testing.

The reservoir in the subject infected with HIV-1 subtype B harbored 6 proviruses across the genome, with four of these provisionally interpreted as expanded clones (Fig 6A). The distribution of clonal expansion and the genes targeted by these clones are shown in Fig 6B. Despite undetectable VL in the circulation, HIV SMRTcap detected circulating provirus-containing cells, including 2 major clonal clusters on chromosomes 5 and 17. In the case of chr17, these integrations occurred in an area where two genes overlapped, *ENSG000002640958* and *SMARCE1*. Our analysis pipeline identified two distinct clonal clusters in the case of the Chr17 integration which were only 5 bp apart. These were single-flanked reads captures, so one clonal cluster represents the expansion of a clone that captured only the 5' integrated provirus, while the other cluster captured only the 3' end of the integrated provirus. Although likely the same clone, we are still currently evaluating the addition of assembling 5' and 3' reads from the same integration site(s) in the HIV SMRTcap analysis pipeline to reconstruct more "dual-flanked" proviral genomes.

With the longitudinal samples collected in Uganda, we aimed to both investigate our ability to characterize HIV-1 subtype A and A/D recombinant reservoirs in an ART-suppressed context and evaluate HIV-1 reservoir dynamics over time. First, we evaluated the composition of the reservoir in Donor 13, for whom we had access to PBMC collected at four timepoints at 12-, 14-, 15- and 17 -years after initiating ART (Fig 6C). HIV SMRTcap identified HIV-1 proviral-containing templates at all timepoints tested, although all were defective. When clonal expansion as a function of integration site was evaluated, the predominant integration site(s) were found in chromosome 3, specifically in the *CCR3* chemokine receptor gene; this clone was identified across all timepoints evaluated (Fig 6D, black arrow). Interestingly, the long non-coding RNA annotated in the UCSC Genome Browser as ENSG00000288724, which is adjacent to *CCR3* (located ~18kb downstream on chr3), also contained an integrated, expanded clone, suggesting that integration in this region of the genome likely helped to convey cell survival over time.

We also evaluated Donor 17 for HIV-1 reservoir composition, clonal expansion, and clonal maintenance throughout ART-suppressed infection (Fig 7A and 7B). This individual was infected with HIV-1 subtype A1, and samples were examined from 9-, 11-, 12-, and 14-years after ART initiation. In contrast to the Donor 13, Donor 17 demonstrated a more diverse reservoir, with smaller clones distributed across a variable collection of integration sites. Only one of these integration sites, in *BACH2*, a gene involved in the primary adaptive immune response, appeared at more than one timepoint (Fig 7B, black arrow) [24,56–58]. Other integrations appeared in immune-activation and interferon-stimulated genes, including *TAP2, ISG15,* and *STAT5*, the latter of which also representing a known "hotspot" gene frequently harboring HIV-1 in expanded clone*s* in other clinical cohorts [24,56–58]. In all cases, regardless of subtype, the viruses detected were internally deleted or indeterminate/truncated. Together, these data demonstrate the ability of HIV SMRTcap to both capture continuity in reservoir composition to evaluate the dynamics of specific clones over time, as well as the breadth of HIV-1 reservoir diversity.

## Comparison to other standard reservoir characterization methods

A unique feature of HIV SMRTcap is that enrichment is not based on a single conserved site within the HIV-1 genome. In fact, the HIV-1-specific probe pool has oligos tiled across the entirety of the genome but only requires ~10–25% tiling

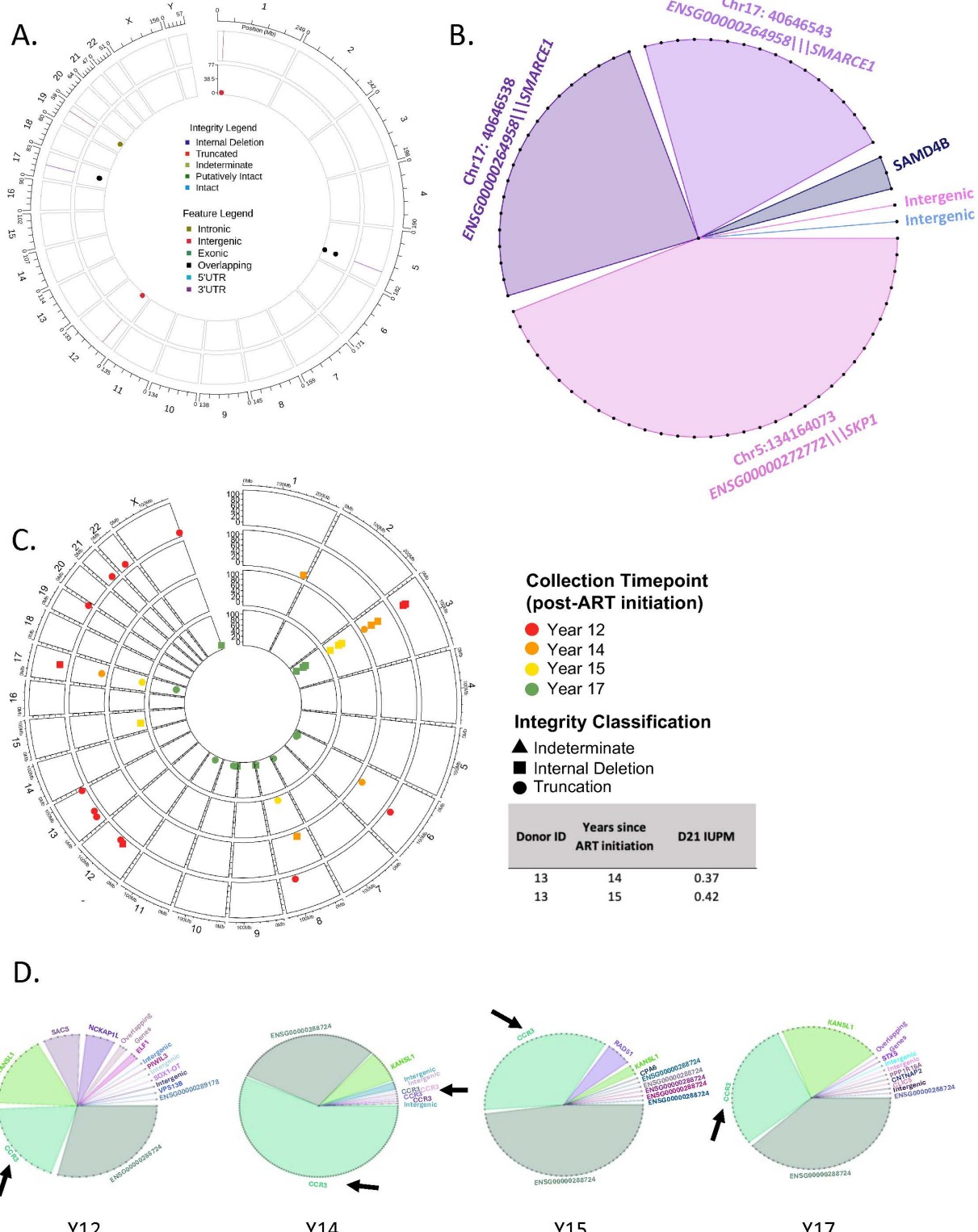

**Fig 6. HIV SMRTcap characterization of ART-suppressed individual across HIV-1 subtypes. (A)** Circos plot representation of HIV-1 reservoir in PBMCs collected from an individual (P4) infected with HIV-1 subtype B virus. **(B)** Clonality analysis of proviruses found in subject P4. Each dot represents a unique provirus detected by HIV SMRTcap. Connected dots filled with the same color represent proviruses belonging to a single expanded

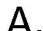

clone sharing the indicated integration. **(C)** Longitudinal proviral profiles from PBMC samples collected from Donor 13 infected with recombinant HIV-1 subtype A1/D. Each layer represents unique integration sites detected at 12 years (red), 14 years (orange), 15 years (yellow), and 17 years (green) after ART initiation. QVOA-based infectious units per million (IUPM) outgrowth values (at day 21) for the indicated, matched time points in Donor 13 are presented in the companion table and were sourced from Ferreira *et al*. **(D)** Clonality analysis for Donor 13 at each time point, including respective integration sites with persistent clones shown in the same color across time point graphs. Black arrows represent relevant integrations at CCR3.

## A.

| Donor ID | Years since ART initiation | D21 IUPM |
|---|---|---|
| 17 | 9 | 0.76 |
| 17 | 11 | 40.52 |
| 17 | 12 | 2.27 |

## B.

Y9    Y11    Y12    Y14

**Fig 7. HIV SMRTcap characterization of ART-suppressed individual ("Donor 17") infected with recombinant HIV subtype A/D. (A)** Longitudinal proviral profiles from PBMC samples collected from Donor 17, infected with HIV-1 subtype A at 9 (red), 11 (orange), 12 (yellow), and 14 (green) years after ART initiation. QVOA IUPM values for the indicated, matched time points in Donor 17 are presented in the companion table and were sourced from Ferreira *et al*. **(B)** Clonality graphs corresponding to each time point for Donor 17. Only shared integration sites share colors across time point. Black arrows represent integrations in BACH2.

coverage to capture a ~ 10–15kb gDNA fragment. Thus, if there is a high level of sequence divergence between the capture probes and primary sample, enrichment should still occur. To finalize HIV SMRTcap development, we aimed to use this relatively unbiased approach to evaluate how HIV SMRTcap compared to or informed on other methods commonly used for HIV-1 reservoir characterization.

As HIV SMRTcap is neither LTR-based, nor LTR-specific, we first used HIV SMRTcap to investigate efficiency of LTR-primed integration site identification, commonly performed when using LM-PCR. In primary HIV-1 subtype B-infected PBMC from ART-suppressed subject P4 (detailed above) we defined an LTR population that differed widely in length (Fig 8A). When evaluating how many of these LTRs would be detectable by standard LM-PCR primers using a histogram analysis, we noted that many LTRs present in the sample were shorter than where these primers would bind, suggesting that the proviral genomes associated with these LTRs may not be captured by standard protocols, which may account for some of the differences observed between the integration sites mapped via LM-PCR versus HIV SMRTcap (Fig 2) [28].

We next compared HIV SMRTcap to the intact proviral detection assay (IPDA), a digital-droplet PCR-based quantitative method to identify intact and proviral genomes in the HIV-1 reservoir. While the original assay was tailored to HIV-1 subtype B, a recent publication has adapted IPDA for HIV-1 subtypes A1, D, and recombinants [32,40]. A few of the samples used for HIV SMRTcap validation had previously been assessed by IPDA, allowing for direct comparison of methods on the same samples (Fig 8B and 8C). In both the HIV-1 subtype B sample P4 (Fig 8B) and the HIV-1 A1 infected sample from Donor 17 (Fig 8C), the IPDA results detected a small but quantifiable population of intact proviruses (green), although most of the detected proviral genomes were internally deleted/defective (red). In contrast, all HIV SMRTcap characterized proviral genomes from both samples were identified as internally deleted/defective. To understand what features contributed to the discrepancy in these results, we further investigated the HIV SMRTcap proviral coverage gene segment-by-gene segment for both samples (Fig 8D). IDPA uses primers that amplify fragments between the 5'-proximal Ψ (*psi*) packaging signal and *gag*, as well as within the 3' end of *env*; the combination of detectable bands in these two targets have been shown to be strongly predictive of intact proviral genomes [32]. Our examination found that, in both samples, deletions of the accessory genes *vif, vpr, and vpu* were responsible for the HIV SMRTcap-specific internal deletion calls – deletions not detectable by IPDA. Despite this, IPDA is not used to explicitly capture and characterize proviral intactness but to quantify proviruses that are likely to be replication competent. While the noted HIV SMRTcap-detected deletions in accessory genes may not render a virus replication-incompetent or completely defective in the same way that deletions in *gag* or *env* may, altered accessory gene expression can directly impact viral pathogenesis by altering susceptibility to antiviral factors key to immune system activity.

Lastly, Donor 17 had also undergone longitudinal full-length integrated proviral sequencing (FLIP-seq) profiling for a previous study [40]. We used HIV SMRTcap data, collected from the same time point to examine the phylogenetic relatedness between the two data types (Fig 8E). As expected, multiple FLIPseq clusters were intercalated within the larger collection of HIV SMRTcap sequences, showing that the two data types were closely related and did not contain technical artifacts that would distinguish them completely. We did observe one FLIPseq sequence cluster that was separated at a more distal branch than the rest of the data; these data contained hypermutated sequences, which may not be resolved with the HIV SMRTcap assay. Together, these cross-validation data demonstrate that the HIV SMRTcap pipeline provides comprehensive resolution of HIV-1 persistent reservoirs compared to and in combination with other widely used HIV reservoir characterization methods.

## Discussion

In this study, we introduce HIV SMRTcap, a novel single-molecule sequencing assay capable of simultaneously resolving integration sites and assessing proviral integrity of HIV-1 persistent reservoirs with single-molecule resolution. The ability to better characterize the reservoir has advanced the HIV-1 cure field, and HIV SMRTcap decreases the variety of assays needed and reliance on indirect estimations to define HIV-1 reservoir composition [24]. HIV SMRTcap thereby provides a novel approach to capturing a comprehensive and unbiased snapshot of the proviral landscape. HIV SMRTcap can serve as either a complementary assay to more general reservoir assessment tests like IPDA, or as an initial survey test prior to more labor-intensive, single cell-based evaluations or viral outgrowth assays.

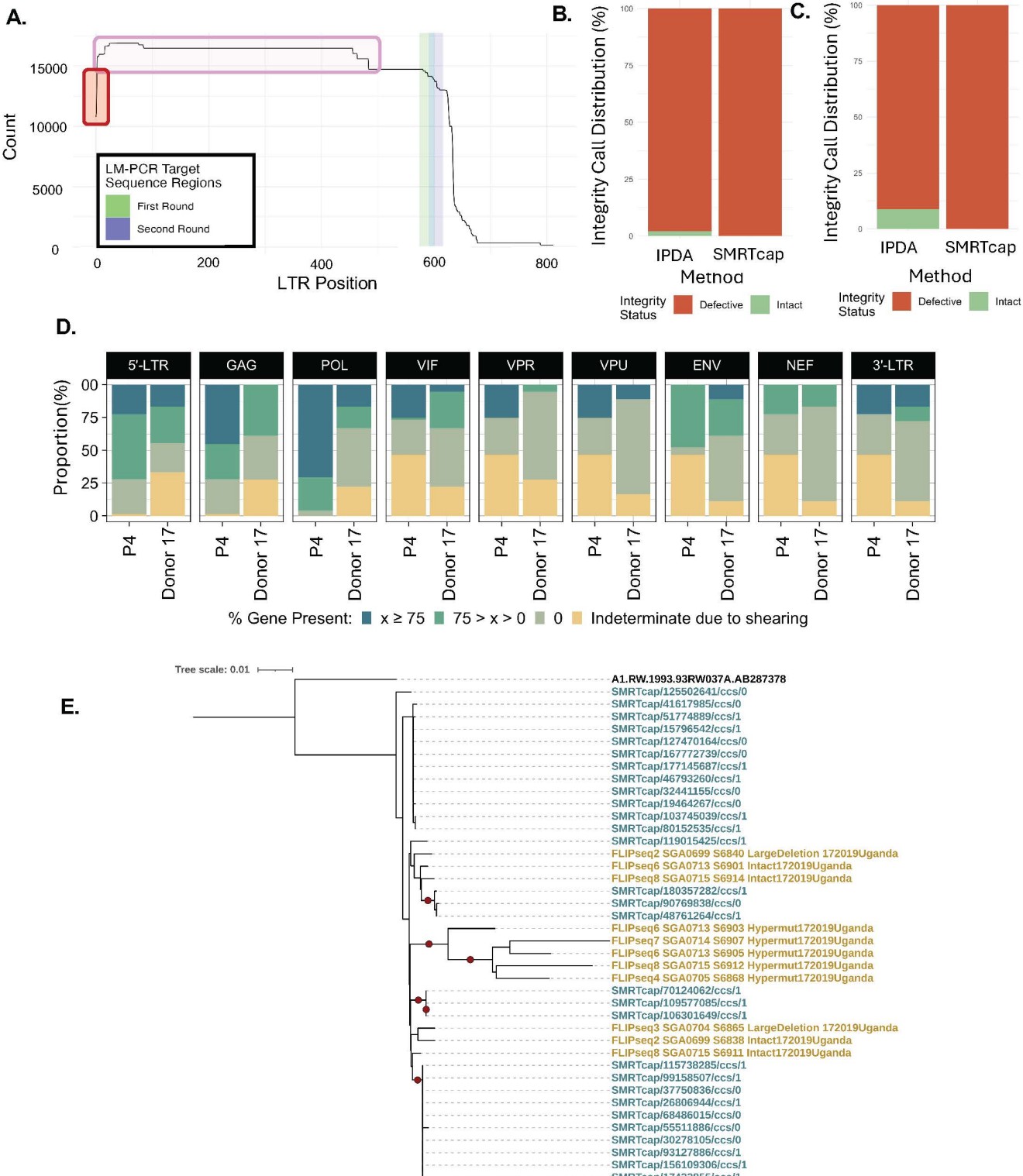

**Fig 8. HIV SMRTcap comparison to previous methods. (A)** 5' LTR length histogram of proviruses detected by HIV SMRTcap in PBMCs collected from subject P4. The red box highlights proviruses missing the 5' LTR. LM-PCR nested primer binding regions clustered around x-axis position 600 are shown in green and blue bars. The pink box identifies proviral sequences harboring truncated LTRs which no longer contain the LM-PCR primer binding

sites and may escape LM-PCR detection. **(B-C)** Stacked bar graphs show proportions of intact proviruses detected by IPDA versus HIV SMRTcap in subject P4 and Donor 17, respectively. **(D)** Gene-by-gene summary of viral content in subjects P4 and Donor 17, showing deletions in accessory genes (*vif, vpr, vpu, nef*). **(E)** Phylogenetic analysis of unique HiFi sequencing reads detected by HIV SMRTcap (blue) and FLIPseq (yellow) methods detected in Donor 17. The red dots indicate a bootstrap value > 0.9 (range: 0-1).

Here we demonstrate that HIV SMRTcap can be used with relatively small quantities of genomic DNA (2 x 10$^6$ PMBC or 2µg gDNA) through targeted enrichment and amplification, although performance is proviral copy number dependent. The strength of the approach comes from the combination of long-fragment based capture and single-molecule sequencing. The first ensures that even partial probe tiling – measured down to 10% coverage – will capture the entirety of the gDNA fragment; the second allows for resolution of all fragments captured as highly accurate, single molecules providing complete integration site and proviral resolution without bioinformatic imputation or read reconstruction. This avoids pitfalls associated with barcode crossover or assembly. Moreover, as our capture probes were designed to tile across the entirety of multiple HIV-1 subtypes, our method is LTR-independent, subtype-agnostic, and works with any extracted HMW gDNA, making it optimal for circulating cell or tissue-based reservoir characterization. Lastly, the high sensitivity of HIV SMRTcap allows for the description of HIV-1 proviral reservoirs in ART-suppressed individuals, which is critical for reservoir characterization with today's treatment regimens.

Like all reservoir assessing assays to date, HIV SMRTcap has several limitations. First, HIV SMRTcap is semi-quantitative. Second, like all comparable methods, HIV SMRTcap is input template limited, which could be an issue for low proviral copy number samples such as those found in elite controllers and individuals with suppressed viremia for decades. In these types of cases, initially undetectable samples could theoretically be iteratively characterized with replicate preparations and deeper sequencing than what has been presented here. Detection of extremely rare templates may be further exacerbated by the subsampling of original input material that occurs with each step of the molecular protocol. If resolved as only a single-flanked (5' or 3' only) provirus due to shearing artifacts or only as a single template with this integration site, this could lead to erroneous clonal identification, functional underestimation of clone size, exaggeration of clonal diversity, or hinderance in identifying local integration hotspots. HIV SMRTcap is likely also limited in its ability to capture highly hypermutated proviruses, as the extensive sequence divergence would be expected to reduce overall capture efficiency and recovery of these variants. This is currently being systematically evaluated in the laboratory; however, as hypermutated viruses are likely not to be replication competent, their absence in these datasets was not considered a critical fault.

Notably, HIV-1 reservoir characterization methods such as IPDA predict intact provirus defined by presence of limited primer binding sites. The discrepancy in detection of "intact" proviruses between pre-existing methods and HIV SMRTcap likely represents the more stringent definition of intactness in HIV SMRTcap data, enabled by nucleotide-level resolution of integrated proviral content. However, a provirus may retain small deletions in accessory genes as observed here and be categorized as "internally deleted" by HIV SMRTcap but still maintain inducibility and replication competence *in vivo*. In such a situation, functional validation using assays such as the quantitative viral outgrowth assay (QVOA) could be used to evaluate the differences between "internally deleted" and truly "defective" proviral genomes. In fact, parallel evaluation of samples by HIV SMRTcap and QVOA did indicate that replication-competent, inducible proviruses were present in these individuals at the time points indicated in Fig 6D. However, the frequency of these infectious units was extremely low (< 1 IUPM), and may not have been represented as intact genomes in the HIV SMRTcap data because either (i) they were under the limit of detection for HIV SMRTcap, which was performed using total PBMC gDNA input, compared to pre-sorted resting CD4 + T cells in QVOA; or (ii) these replication-competent proviruses still contained small deletions in accessory genes and may be represented within the HIV SMRTcap data, but these deletions did not impact inducibility or infectivity. These possibilities underscore the importance of functional, orthogonal validation of all genomics-based HIV reservoir characterization methods.

Given the known challenges of HIV-1 reservoir characterization, including (but not limited to) extremely low frequency during viral suppression, tissue-specificity and compartmentalization, limited sample availability, and highly divergent HIV-1 subtype-specific features, HIV SMRTcap provides a novel, innovative approach to complement the existing assays available. This high-throughput, comprehensive pipeline can be universally applied across populations without requiring *a priori* knowledge of HIV-1 subtype and has been designed to be sensitive, making it applicable to a variety of sample types. HIV SMRTcap holds significant promise for enhancing our basic understanding of HIV-1 persistence and pathogenesis, including anatomical sanctuaries that may preferentially drive viral evolution or reseed infection upon therapy interruption, investigate how integration sites or their genomic contexts impact proviral genome integrity and persistence, and/or offer insight into how reservoir dynamics affect recombinant HIV formation in the contexts of co- or super-infection. Additionally, HIV SMRTcap will provide a sensitive, accurate pipeline to evaluate residual reservoir composition when testing next-generation HIV-1 cure strategies across worldwide subtypes.

## Materials and methods

### Ethics statement

Ethical Approvals and Participant Recruitment: The participant (WWH-B31) was recruited from Whitman Walker Health (Washington, D.C.) under a protocol approved by the George Washington University Institutional Review Board. Secondary use of samples was approved by the Weill Cornell Medicine Institutional Review Board. The participant provided written informed consent.

### Samples used and collection details

Samples used for the demonstration of HIV SMRTcap capability are summarized in Table 1. Details of their generation, collection, and inclusion into this study are provided below.

**In vitro samples.**

8e5 cells: 8e5 cells were obtained from the NIH HIV/AIDS reagent repository program (BEI Resources Repository). 8e5 cells were maintained in RPMI media supplemented with 10% Fetal Bovine Serum, 2.5% HEPES buffer, 1% penicillin/streptomycin, and 1% L-Glutamine and passaged routinely. Early passage aliquots were resuspended in cell freezing buffer and snap frozen prior to storage in liquid nitrogen. When needed, 8e5 cells were thawed and HMW gDNA was extracted using the MagAttract kit (Qiagen #67563), as instructed by the manufacturer. Limit of detection experiments were performed with early passage (passage #2) cell aliquots. Cells were then passaged extensively and frozen down at the passage numbers indicated in the experiments shown. Cryopreserved cells were used for the multi-passage experiments, and HIV SMRTcap was performed on all four differently timed passage aliquots in parallel.

SupT1 cells and *in vitro* viral infection: SupT1 cells (negative control) and SupT1 cells expressing CXCR4 and/or CCR5 (SupT1-R5 or SupT1-X4, respectively) were used for *in vitro* infection by individual HIV-1 subtypes. Infectious clones of viruses that represent all four major subtypes were obtained from the NIH HIV/AIDS reagent repository, including: 92UG_029 (subtype A), 89BZ_167 (subtype B), 93MW_965 (subtype C), and 93UG_065 (subtype D). HIV-1-infected SupT1-R5 or SupT1-X4 cell pellets were collected during active infection, and HMW gDNA was extracted using the GenomicTip kit (Qiagen #10223) for downstream HIV SMRTcap processing.

**Primary samples.**

Human post-mortem tissues: Brain (frontal lobe and basal ganglia) and heart tissues were obtained and characterized by the Manhattan HIV Brain Bank (member of the National NeuroHIV Tissue Consortium, NNTC), using protocols and consents approved by the Icahn School of Medicine at Mount Sinai Institutional Review Board. The MHBB conducts a longitudinal, observational study of people with and at risk for HIV-1 who are willing to serve as organ donors upon demise; as part of this study, plasma viral loads and T-cell enumerations are routinely assayed in CLIA-certified hospital

laboratories, and ART utilization is documented by medical interview. At the time of autopsy donation, tissues are rapidly frozen between aluminum plates at minus 85°C and stored at that temperature until use. Routine histology is assessed as previously described [59].

HIV-1 subtype B-infected PBMC:    HIV-1-infected individuals were prospectively followed as part of a multicenter, nation-wide observational study and clinical trial examining solid organ transplantation of individuals living with HIV-1 from donors with or without HIV-1 [51,55,60]. PBMC were collected at time of transplant and at multiple time points after transplant (approximately 13, 26, 52, 104, and 156 weeks after transplant, as indicated) as previously described [61]. During transplantation, recipients underwent either lymphocyte-depleting (e.g., ATG or alemtuzumab) or non-depleting (e.g., basiliximab) induction therapies [62]. The sample processed by HIV SMRTcap, "P4", and associated metadata are indicated in Table 1. The trial and current study were approved by the Johns Hopkins University School of Medicine Institutional Review Board. All transplant participants provided written informed consent.

HIV-1 subtype A and A1/D-infected PBMC:    Cells were obtained from Ugandans living with HIV-1 who were virally suppressed and participating in the Rakai HIV-1 Latency study, which has been described in detail previously [52–54]. Briefly, Ugandans living with HIV-1 who were virally suppressed and initiated ART at least one-year prior were eligible to enroll. Large blood draws were taken annually, PBMC were isolated and stored, and were used to perform quantitative viral outgrowth assays. In addition, PBMC samples from an unmatched timepoint for Donor 13 were also previously sequenced using FLIP-seq to examine near full-length proviral sequences as described previously [40] and were used for HIV SMRTcap comparative analyses. The study was approved by the National Institute of Allergy and Infectious Diseases (National Institutes of Health) and the Uganda Virus Research Institute, Uganda National Council for Science and Technology. All participants provided written informed consent. Participants were provided an honorarium of 20,000 Ugandan Shillings (~5 US dollars) per completed study visit. In addition, participants were reimbursed between 8000 and 50,000 Uganda shillings (~2–12.5 US dollars) per visit for transportation costs depending on distance and economic conditions. Total reimbursement may be up to 70,000 Uganda shillings (~17.5 US dollars) per visit.

Participant-derived xenograft mouse model (PDX) spleen cells:    PBMC Isolation and CD4 + T Cell Enrichment: Peripheral blood mononuclear cells (PBMCs) were isolated from leukapheresis samples and enriched for CD4 + T cells as previously described [63]. Briefly, leukapheresis processing involved Ficoll-Paque density gradient centrifugation (GE Healthcare), followed by CD4 + T cell enrichment using the EasySep Human Memory CD4 + T Cell Enrichment Kit (STEMCELL Technologies, catalog #19157). Enriched cells were resuspended in RPMI 1640 medium (Gibco) supplemented with 10% fetal bovine serum (FBS), 1% penicillin/streptomycin, and 1% HEPES, then maintained at 37°C until use.

Xenograft Mouse Model: Six-week-old NOD.Cg-Prkdcscid Il2rgtm1Wjl/SzJ (NSG) mice (The Jackson Laboratory, stock #005557) were engrafted with $5 \times 10^6$ memory CD4 + T cells via tail vein injection as detailed in [63]. Five weeks following engraftment, mice were infected via intraperitoneal injection with high titer doses (10,000 $TCID_{50}$ diluted in 100mL total Hanks' Balanced Salt Solution) of $HIV_{JR-CSF}$, a subtype B virus. Mice were housed under specific pathogen-free conditions with daily health monitoring by trained technicians, in compliance with AAALAC International guidelines.

Tissue Harvest and Processing: Eight weeks post-infection, mice were euthanized by $CO_2$ asphyxiation followed by cervical dislocation. Spleens were harvested, mechanically dissociated using a 10 mL syringe plunger and incubated for 10 minutes at 37°C in RPMI 1640 (supplemented as above) with 25 U/µL DNase I (Sigma-Aldrich, Cat. # D5025). Cell suspensions were filtered through a 70 µm strainer, resuspended in freezing medium (90% FBS + 10% DMSO), and stored at −80°C prior to shipment on dry ice to the University of Louisville.

All procedures were approved by the Weill Cornell Medical College Institutional Animal Care and Use Committee (IACUC protocol #2018–0027).

**SMRTcap enrichment of HIV-1 proviral DNA.**

Quality control and inclusion of HMW gDNA:    gDNA was extracted from samples using the MagAttract HMW DNA kit (QIAGEN #67563) following the manufacturer's instructions. Briefly, samples were lysed, cellular proteins digested with

Proteinase K, and any contaminating RNA digested by RNase A. Purity, quality, and quantity of gDNA extracted from all sources was evaluated by the Nanodrop 2000 (ThermoFisher), TapeStation 4150 Genomic DNA Kit (Agilent), and Qubit 4 fluorometer dsDNA High Sensitivity 1X reagents (ThermoFisher), respectively. HMW gDNA > 20 kb was taken forward for mechanical shearing and further sample processing. Samples with gDNA fragments between 10–20 kb were used for HIV SMRTcap enrichment without further shearing. Samples with gDNA fragments < 10 kb suggested highly degraded material that would not allow for simultaneous resolution of HIV-1 proviral sequences and integration sites and were excluded from further processing.

Shearing of HMW gDNA: Two and a half micrograms (2.5 µg) of HMW gDNA per sample was mechanically sheared to 10–15 kb fragments using g-TUBE (Covaris, #520079). Centrifugation was performed at 4100 rpm using an Eppendorf 5242R microcentrifuge. The sheared gDNA was then purified and concentrated with AMPure PB beads (Pacific Biosciences #100-265-900) using a 0.5X vol:vol ratio.

Enrichment of HIV-1 proviral-containing gDNA fragments: Capture and enrichment of HIV-1-containing gDNA fragments was performed according to the "Long Read Library Preparation and Standard Hyb v2 Enrichment" protocol from Twist Bioscience according to the manufacturer's guidelines with only minor modifications. Briefly, sheared gDNA underwent end repair and A-tailing, followed by ligation of an annealed barcoded universal adapter, which is used both for downstream amplification and to enable sample multiplexing. We did not use the Twist Universal Adapters, but the Barcoded Adapter Oligo Pairs designed by Pacific Biosciences. Following adapter ligation, fragments were purified using 0.5X vol:vol AMPure PB beads.

Pre-capture amplification was performed using forward and reverse primers specific for the barcoded linear adapters and one microgram (1 µg) sheared gDNA. Distinct from the standard protocol conditions, we amplified using PrimSTAR GXL Polymerase (Takara, #R050B), which was more capable of robust amplification of longer fragment sizes. Following amplification, fragments were again assessed for quality and quantity (QC), using the TapeStation 4150 (Agilent) and Qubit fluorometer 4 (ThermoFisher). Size selection was performed using the BluePippin system (Sage Science) to remove all fragments < 8 kb prior to probe hybridization. QC performed after size selection determined the fragment sizes and concentrations needed for equimolar pooling of samples, if performed. Up to four samples were used per pool, each ligated to a distinct universal barcode to ensure efficient sample identification post-sequencing. Samples were equimolar pooled prior to enrichment with a total mass of up to 4 µg size-selected, pooled amplified gDNA input for the oligo capture step.

Barcoded pooled libraries were prepared for enrichment by desiccation in a vacuum concentrator with no heat and then reconstituted in a small volume of Blocker Solution (Twist #100774) followed by mixing with Hybridization Mix (Twist #100528) and our custom, pan-subtype HIV-1 oligo probe pool. The mix was covered with the Hybridization Enhancer (Twist #100986) and allowed to hybridize at 70°C for 16–18 hours. Hybridized fragments were isolated by incubating the pool with M-270 Dynabeads (Invitrogen #65305) for 30 minutes at room temperature. Multiple wash steps with Wash Buffer I and II (Twist #100589 and #100590, respectively) removed non-hybridized material and low-affinity bound off-target fragments. The captured library was eluted from the beads with a mixture of water, 0.2N NaOH, and 200 mM Tris-HCI. Post-capture amplification of the enriched material was performed using the KOD Xtreme Hot Start Polymerase (Millipore Sigma #71975-M) across two replicate reactions, as specified in the standard protocol. Final captured libraries were assessed for quantity and quality using the Tape Station 4150 and Qubit 4 fluorometer, respectively.

## Pan-subtype HIV-1 oligonucleotide enrichment probe pool

Oligonucleotide probes (120-mers) specific for the HIV-1 genome were designed by Twist Bioscience using a multi-sequence alignment of 23 full-length HIV-1 viruses across 11 distinct HIV-1 subtypes and recombinants (Table 3). The proprietary algorithm (Twist Bioscience) used for design reduces overrepresentation of oligos in conserved regions and provides additional coverage in regions with excessive diversity across references (e.g., HIV-1 *env*). This HIV-1 pan-subtype oligo pool was used for all experiments presented.

**Table 3. Full-length viral genome references used for HIV SMRTcap capture probe design.**

| Isolate Name | HIV-1 Subtype | GenBank Accession |
|---|---|---|
| 92RW008_A04 | A1 | AB253421 |
| UG031-A1 | A1 | AB098330 |
| 93RW037A | A1 | AB287378 |
| 92UG037_A40 | A1 | AB253429 |
| PBS1195 | A2 | MH705163 |
| HXB2 | B | K03455 |
| Ba-L | B | AB221005 |
| JPDR388 | B | AB289589 |
| JPDR6089 | B | AB286955 |
| 93IN01 | C | AB023804 |
| 02ZM112 | C | AB254144 |
| ZAM18 | C | AB485645 |
| 96BW06H51 | C | AF290027 |
| SE365 | D | AB485649 |
| UG270 | D | AB485650 |
| 92UG001 | D | AJ320484 |
| p190049 | D | JX236668 |
| PRD320-14F45 | F1 | AB485658 |
| A1699 | F2 | MH705144 |
| DRCBL | G | AF084936 |
| LA19KoSa | H | KU168273 |
| LA26DiAn | J | KU168280 |
| 93JP-NH2.5T | AE | AB070352 |

## SMRTbell library construction

The enriched sample pool(s) were used as input for SMRTbell library preparation with the SMRTbell Express Template Kit 2.0 (PacBio #100-938-900) following the manufacturer's instructions with minor modifications. Briefly, the enriched sample pool underwent DNA damage repair for two hours, followed by end repair and A-tailing. Samples were then used for SMRTbell adapter ligation for 1 hour followed by a ligase-killing step in which the mix was 65°C for 10 minutes and returned to 4°C. Nuclease treatment of the SMRTbell libraries was performed using the SMRTbell enzyme clean up kit (PacBio #101-746-400) as written, but with an incubation for two hours, to eliminate damaged and/or unligated templates that would interfere with productive loading of the library on the sequencing system. Final HIV SMRTcap SMRTbell libraries were purified using 0.6X AMPure PB beads.

## Sequencing on the Sequel IIe and/or Revio systems

Almost all libraries presented here were sequenced on the Sequel IIe system (Pacific Biosciences, Table 2). To prepare libraries for sequencing, HIV SMRTcap SMRTbell libraries were initially annealed to a sequencing primer (v4) prior to binding to the sequencing polymerase (v2.0). Unbound polymerase was washed away, and polymerase-bound SMRTbells were loaded onto the Sequel IIe system as specified by the manufacturer using the ">3kb amplicon" loading parameters and protocol. For the few samples sequenced on the Revio system, library preparation for sequencing was similar; however, in these cases we used the SPRQ sequencing primer and polymerase, with the "Target Enrichment" loading parameters and protocol within SMRTLink v13. For both systems, data were collected using 30-hour movies and generation of

high fidelity ("HiFi", accuracy > 99.9%) reads were generated on-instrument. Demultiplexing of pooled libraries was performed off-instrument, and these demultiplexed, HiFi read files (FASTQ) were used for downstream data processing.

**SMRTcap analyses.** All data were analyzed by a three-step pipeline to: (i) identify the sequencing reads that contained HIV-1 material; (ii) define integration sites and enumerate expanded clones; and (iii) characterize proviral integrity per molecule. Details for each step are provided below.

*Identifying HIV-1 integrations and host flanking sequences.* HIV SMRTcap CCS reads at a Phred Quality score of 30, or 99.9% accurate, or higher were determined to be High Fidelity (HiFi) reads. HiFi reads were mapped to the HIV-1 reference genome database listed in Table 3 using minimap2 v2.24-r1122 with the parameters `-t 16 -m 0 -Y -ax map-pb` to identify reads with HIV-1 integrations [64]. If the subtype was known, references were limited to the matched genome; otherwise, all references were used. Reads were then filtered to exclude secondary alignments and unmapped reads using samtools view v1.16.1 with the parameters `-h -S -F 260` [65]. Reads were further filtered to exclude secondary alignments (`-F 256`), unmapped reads (`-F 4`) and supplementary alignments (`-F 2048`) using the samtools view command with the `-F 2308` parameter. Additional parameters `-h` and `-S` were included to retain the header and specify the input format as SAM. The filtered reads were then hard masked for the regions containing hits to the HIV-1 reference and iteratively searched again using minimap2 with the `parameters -t 16 -Y -p 0 -N 10000 -ax map-pb` until no additional HIV-1 hits were found. At the end of this iteration, all regions of the reads containing an HIV-1 hit were hard masked. The hard masked reads were then searched against the primary GRCh38 human genome assembly to identify insertion site flanks using minimap2 and the parameters `-t 16 -Y -p 0 -N 10000 -ax map-pb` which were then filtered to remove supplementary alignments, secondary alignments, and unmapped reads with samtools view and the parameters `-h -S -F 2308`. The resulting reads were then processed using a custom python script combine_hiv.py that parses the sam alignment files to produce a tab-delimited file containing information about the HIV-1 sequence hit, and the left and right human flanking sequences.

*Mapping human flanking sequences and assigning clonal expansion events* The HIV-1-mapped summary.csv file was imported into the R (version 4.3.1) pipeline via openxlsx (version 4.2.6.1). Data was cleaned through use of tidyr (version 1.3.1), dplyr (version 1.1.4), purrr (version 1.02), stringr (version 1.5.1), and stringi (version 1.8.4) [66–70]. Quality control of reads was performed through examination of human genome flanking coordinates to identify mis-mapped reads or paired integration sites existing at disparate genomic loci (indicative of PCR artifact). Reads that did not pass these quality parameters were moved to a "low confidence" bin, while those that did pass were considered "high confidence". Clonal expansion events were defined as those molecules that contained identical integration sites, but unique shearing end coordinates (schematized in Fig 1). Integration sites were considered to have high confidence mapping if they were within a 10 bp window of each other (± 5 bp up- or downstream) to account for polymerase slippage during amplification. 10 bp was chosen as the maximal spread with which the 5' and 3' integration sites could differ because of the well-described polymerase slippage of up to 5 bp and existing technical difficulty in many sequencing platforms with homopolymeric sequence identity [71,72]. Similarly, integration coordinates between different reads that overlapped (e.g., chr1:300,600,428–300,600,432 and chr1:300,600,430–300,600,433) were considered clones of the same initial HIV integration event with discrepancies assumed to be from the sequencing process. In parallel, PCR duplications were identified as molecules that shared both integration sites and shearing coordinates. The coordinate, clonal expansion, and PCR data were mapped to genes (including identification of sub-genomic features), intergenic space, repeat elements, and ENCODE features (e.g., CpG Islands, promoters, enhancers) as pulled from the UCSC Genome Browser via BiocManager (version 1.30.24), GenomicRanges(version 1.52.1) and plyranges (1.20.0). All associated R software package dependencies were used with default settings.

*Gene intactness.* A custom perl script `findHIVSIVGeneRegions` was used to determine the intactness of the overall integration. This script measured the intactness of the 5' LTR, 3'LTR as well as the *gag*, *pol*, *vif*, *vpr*, *vpu*, *env*, and *nef* genes and genome segments. This was done by comparing the HIV-1 integration sequence against the respective

annotated region from the closest specified HIV-1 reference genome identified during the mapping step using NCBI `blastn` v2.10.0+ [3] with the parameter `-word_size 16` to remove spurious hits. If no hits were found for a particular region, that region was assigned a value of 0. If the length and identity of the match was between 0 and 75% of the reference region, then that gene was assigned a value of 1. Length and sequence identity matches greater than 75% were assigned a value of 2. This parameter (identity of 75% and above qualifying as an "intact" call) was chosen after extensive testing to accommodate within-subtype sequence variation to avoid false deletion calls due to mismapping, and validated by manual curation of up to 100 single molecule reads per sample. A match string of length 9 was produced, corresponding to the matches to the 5' LTR, *gag*, *pol*, *vif*, *vpr*, *vpu*, *env*, *nef*, and 3' LTR regions. A complete intact integration will have a value of "`222222222`". Post-processing of the integration strings was performed to change 0 values at the end to "-" if host flanking sequences were not available, indicating a loss of content to mechanical shearing. For example, a string of "`--2222222`"" indicates that the 5' end of the sequence read contains the *pol* segment, and no additional sequence is available to assess whether the 5' LTR and *gag* regions are present or absent from the corresponding integration (as schematized in Fig 1). All data, including unique sequence read ID, integration sites and feature(s), proviral integrity string, and simulated IPDA results are then compiled in released in a master sheet (.xls) per sample. A sample master sheet detailing the full HIV SMRTcap integration and proviral integrity profiling dataset for the HIV JR-CSF-infected CD4 T cells described in Fig 4C is provided as S1 Table.

**LM-PCR profiling of integration sites.** HIV-1 integration site libraries were generated using ligation-mediated PCR (LM-PCR) as previously described [28,73]. The NEBNext Ultra II FS DNA Library Prep Kit (New England Biolabs), the manufacturer's provided protocol, and the published protocol were employed for this procedure [56]. Genomic DNA (5 μg) was fragmented (35 μl) in a reaction that included 7 μl of NEBNext Ultra II FS reaction buffer and 2 μl of NEBNext Ultra II FS enzyme mix. The reaction was carried out at 37°C for 20 min, followed by enzyme inactivation at 65°C for 30 min. Following spin column purification, three separate ligation reactions were performed (68.6 μl each) at 20°C for 16 h. These reactions included 2.5 μl of 10 μM double-stranded asymmetric linkers containing 3'-T overhangs, 30 μl of NEBNext Ultra Ligation Master Mix, and 1 μl of NEBNext Ligation Enhancer. The ligated DNAs were subsequently purified and subjected to semi-nested PCR to amplify viral-host integration junctions. The amplified PCR products were sequenced (150 bp paired end) at Azenta using the Illumina platform.

### Phylogenetic analysis

**Detection of outgroups in PDX spleen samples.** Consensus sequences were generated for each unique clonal provirus found the HIV SMRTcap data derived from PDX spleen gDNA. These were then aligned to the infecting virus, JR-CSF, reference genome using ViralMSA [64,74]. The alignment output files were analyzed using IQ-TREE 2 (V1.6.12) [75]. We employed standard model selection (-m Test) to determine the best-fitting model of sequence evolution and reconstructed the phylogenetic tree based on the best model [76]. We further performed ultrafast bootstrap analysis (-bb) with 10,000 replicates to measure branch support, and considered bootstrap values >90% as robust indicators of branch separation [77]. Lastly, the interactive Tree of Life software (iTOL V7) was used to visualize the resulting trees, which were presented in an unrooted format due to the absence of timepoint-specific sample collection and comparisons [78].

To validate this separation of viral outgroups, we performed compartmentalization analysis. Specifically, we applied Peter Simmonds's Association Index (AI) analysis in Hypothesis Testing using Phylogenies (HyPhy v2.5.63) to all consensjus sequences [79,80]. We consider an AI score below 0.33 as indicative of a significant compartmentalization event if the separation is also supported by bootstrap values (1,000 replicate) above 90% with 10 times relabeling per sample [81,82]. Our analysis yielded an AI score of 0.00575 with 100% bootstrap support, confirming robust separation of the two clades. To further validate branch support, we applied BOOSTER [83], an alternative method for calculating phylogenetic support values. BOOSTER also confirmed the separation of the two clades, with bootstrap values >70% considered significant.

**Comparison of FLIP-seq and HIV SMRTcap results in matched samples.** PCR duplicates were removed from the FASTA file generated by the HIV SMRTcap custom pipeline using R. The resulting unique sequences from Donor 17 were then combined with sequences obtained via the FLIPseq assay. These merged sequences were aligned to the subtype A1 reference genome using the Python tool ViralMSA [64,74]. The aligned sequences were subsequently analyzed using neighbor-joining (NJ) method with bootstrap of 50, upon which the phylogenetic tree reconstruction was based. Finally, the phylogenetic trees were visualized using the Interactive Tree of Life software (iTOL v7) [78,84].

## Supporting information

**S1 Fig. Low-level coverage of human genome shows little to no endogenous retrovirus enrichment in off-target capture reads.** (A) Alignment of off-target (e.g., non-HIV-containing) reads to a gene-rich region of the human genome, demonstrating an average coverage between 0–16-fold. (B) Alignment of off-target reads to human endogenous retrovirus K, known to be integrated at chr 11 in q13.2 (exact position identified by vertical black dashed line) and surrounding non-viral content. Average coverage of entire fragment is between 0 and 21-fold, similar to that observed in non-lentiviral containing regions of the human genomes.
(TIFF)

**S2 Fig. Phylogeny of viral genomes recovered from PGX mouse reconstituted from human viremic controller.** Phylogenetic analyses of recovered proviral genomes from the spleen of a PGX mouse reconstituted with lymphocytes from a human viremic controller demonstrates three statistically distinct populations. The majority of sequences (black) are very similar whereas the statistically significant divergence of two smaller outgroups (blue and pink, respectively) was supported by two independent bootstrap analyses methods (ultrafast and BOOSTER), as well as compartmentalization analysis, which achieved an AI = 0.00575, confirming robust population separation.
(PDF)

**S1 Table. Sample mastersheet for HIV SMRTcap data.** This table provides an example report of the data provided by HIV SMRTcap following computational pipeline analyses, including integration sites, clonality, and proviral integrity values. The data presented in this table was derived from the PDX mouse model infected with JR-CSF.
(XLSX)

**S2 Table. Complete viral load history of viremic controller used in PDX model.** This table includes all viral load measurements taken from the viremic controller used in the PDX mouse model, highlighted in Fig 4 and S1 Table.
(DOCX)

## Acknowledgments

The authors want to extend deep gratitude to the University of Louisville Sequencing Technology Center staff for close partnership that ensured robust data production for this new, off-specification long-read protocol. The authors would also like to thank the participants and staff of the HOPE in Action study, the Rakai Latency Cohort, and the National Neuro-AIDS Tissue Consortium (Center ID: 75N95023C00015). "SMRT" is a registered trademark of Pacific Biosciences, Inc.

## Author contributions

**Conceptualization:** Ghazal Sadri, James Powell, Peter Warburton, Gintaras Deikus, Tina Han, Aaron A.R. Tobian, Alan N. Engelman, Robert Sebra, Susan Morgello, Andrew D Redd, David Sachs, Eric Rouchka, Melissa L. Smith.

**Data curation:** Ghazal Sadri, Steven T Nadakal, Parmit K. Singh, Kaitlyn M Shields, R Brad Jones, Alan N. Engelman, Susan Morgello, Andrew D Redd, Eric Rouchka, Melissa L. Smith.

**Formal analysis:** Ghazal Sadri, Steven T Nadakal, William Lauer, Justin Kos, Parmit K. Singh, Erin Elliott, Catherine W. Kaiser, Easton E. Ford, Jessica L Prodger, R Brad Jones, Aaron A.R. Tobian, Alan N. Engelman, Andrew D Redd, David Sachs, Eric Rouchka, Melissa L. Smith.

**Funding acquisition:** Aaron A.R. Tobian, Alan N. Engelman, Robert Sebra, David Sachs, Melissa L. Smith.

**Investigation:** Ghazal Sadri, Steven T Nadakal, William Lauer, Justin Kos, Parmit K. Singh, Erin Elliott, Catherine W. Kaiser, Easton E. Ford, Nadia Richardson, Elizabeth Hudson, Noemi L. Linden, Ali Danesh, James Powell, Peter Warburton, Juan Soto, Matthew Emery, Gintaras Deikus, Tina Han, Alan N. Engelman, Robert Sebra, Susan Morgello, Andrew D Redd, Eric Rouchka, Melissa L. Smith.

**Methodology:** Ghazal Sadri, Steven T Nadakal, William Lauer, Justin Kos, Parmit K. Singh, Erin Elliott, Catherine W. Kaiser, Easton E. Ford, Nadia Richardson, Kaitlyn M Shields, Elizabeth Hudson, Noemi L. Linden, Ali Danesh, James Powell, Peter Warburton, Juan Soto, Matthew Emery, Gintaras Deikus, Guinevere Q Lee, Jessica L Prodger, Tina Han, Alan N. Engelman, Robert Sebra, Susan Morgello, Andrew D Redd, David Sachs, Eric Rouchka, Melissa L. Smith.

**Project administration:** Ghazal Sadri, Parmit K. Singh, Easton E. Ford, Steven J Reynolds, Ronald Galiwango, Jessica L Prodger, Taddeo Kityamuweesi, R Brad Jones, Aaron A.R. Tobian, Alan N. Engelman, Robert Sebra, Susan Morgello, Andrew D Redd, Eric Rouchka, Melissa L. Smith.

**Resources:** Steven T Nadakal, Elizabeth Hudson, Noemi L. Linden, Ali Danesh, Guinevere Q Lee, Susanna L. Lamers, Steven J Reynolds, Ronald Galiwango, Jessica L Prodger, Stephen Tomusange, Taddeo Kityamuweesi, R Brad Jones, Aaron A.R. Tobian, Alan N. Engelman, Robert Sebra, Susan Morgello, Andrew D Redd, Eric Rouchka, Melissa L. Smith.

**Software:** Ghazal Sadri, Steven T Nadakal, Susanna L. Lamers, David Sachs, Eric Rouchka.

**Supervision:** Steven J Reynolds, Ronald Galiwango, Jessica L Prodger, R Brad Jones, Aaron A.R. Tobian, Robert Sebra, Eric Rouchka, Melissa L. Smith.

**Validation:** Ghazal Sadri, Steven T Nadakal, Erin Elliott, Easton E. Ford, Matthew Emery, Gintaras Deikus, Guinevere Q Lee, Ronald Galiwango, Andrew D Redd, David Sachs, Eric Rouchka, Melissa L. Smith.

**Visualization:** Ghazal Sadri, Steven T Nadakal, Nadia Richardson, Susanna L. Lamers, Jessica L Prodger, R Brad Jones, Aaron A.R. Tobian, Robert Sebra, Susan Morgello, Andrew D Redd, Eric Rouchka, Melissa L. Smith.

**Writing – original draft:** Ghazal Sadri, Steven T Nadakal, Andrew D Redd, David Sachs, Eric Rouchka, Melissa L. Smith.

**Writing – review & editing:** Ghazal Sadri, Steven T Nadakal, William Lauer, Parmit K. Singh, Erin Elliott, Catherine W. Kaiser, Easton E. Ford, Nadia Richardson, Elizabeth Hudson, Noemi L. Linden, Peter Warburton, Steven J Reynolds, Jessica L Prodger, Stephen Tomusange, Taddeo Kityamuweesi, Tina Han, R Brad Jones, Aaron A.R. Tobian, Alan N. Engelman, Robert Sebra, Susan Morgello, Eric Rouchka, Melissa L. Smith.

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
