## [Decision Letter · Decision Letter 0]

30 Jun 2025

Development and validation of HIV SMRTcap for the characterization of HIV-1 reservoirs across tissues and subtypesDevelopment and validation of HIV SMRTcap for the characterization of HIV-1 reservoirs across tissues and subtypes

PLOS Pathogens

Dear Dr. Smith,

Thank you for submitting your manuscript to PLOS Pathogens. After careful consideration, we feel that it has merit but does not fully meet PLOS Pathogens's publication criteria as it currently stands. Therefore, we invite you to submit a revised version of the manuscript that addresses the points raised during the review process.

Please submit your revised manuscript within 60 days Aug 29 2025 11:59PM. If you will need more time than this to complete your revisions, please reply to this message or contact the journal office at plospathogens@plos.org. Please include the following items when submitting your revised manuscript:

We look forward to receiving your revised manuscript.

Kind regards,

Jason M. Brenchley

Academic Editor

PLOS Pathogens

Susan Ross

Section Editor

PLOS Pathogens

Sumita Bhaduri-McIntosh

Editor-in-Chief

PLOS Pathogens

orcid.org/0000-0003-2946-9497

Editor-in-Chief

PLOS Pathogens

orcid.org/0000-0002-7699-2064

**Additional Editor Comments:**

Apologies for the long turn around time. One of the reviewers was significantly delayed.

All reviewers found the work interesting, but each raised some concerns which should be addressed.

**Journal Requirements:**

1)We ask that a manuscript source file is provided at Revision. Please upload your manuscript file as a .doc, .docx, .rtf or .tex. If you are providing a .tex file, please upload it under the item type u2018LaTeX Source Fileu2019 and leave your .pdf version as the item type u2018Manuscriptu2019.

2) Please ensure that the Title in your manuscript file and the Title provided in your online submission form are the same.

https://journals.plos.org/plospathogens/s/submission-guidelines#loc-parts-of-a-submission

4) We noticed that you used the phrase 'data not shown' in the manuscript. We do not allow these references, as the PLOS data access policy requires that all data be either published with the manuscript or made available in a publicly accessible database. Please amend the supplementary material to include the referenced data or remove the references.

5) We have noticed that you have cited Tables 1, 2, and 3 in the manuscript file but there are no corresponding tables in the manuscript. Please amend your manuscript to include these tables noting that tables should not be uploaded as individual files.

6) We have noticed that you have uploaded Supporting Information files, but you have not included a list of legends. Please add a full list of legends for your Supporting Information files after the references list.

7) Some material included in your submission may be copyrighted. According to PLOSu2019s copyright policy, authors who use figures or other material (e.g., graphics, clipart, maps) from another author or copyright holder must demonstrate or obtain permission to publish this material under the Creative Commons Attribution 4.0 International (CC BY 4.0) License used by PLOS journals. Please closely review the details of PLOSu2019s copyright requirements here: PLOS Licenses and Copyright. If you need to request permissions from a copyright holder, you may use PLOS's Copyright Content Permission form.

Potential Copyright Issues:

i) Figures 1A, 1B, and 5A. Please confirm whether you drew the images / clip-art within the figure panels by hand. If you did not draw the images, please provide (a) a link to the source of the images or icons and their license / terms of use; or (b) written permission from the copyright holder to publish the images or icons under our CC BY 4.0 license. Alternatively, you may replace the images with open source alternatives. See these open source resources you may use to replace images / clip-art:

8) Please amend your detailed Financial Disclosure statement. This is published with the article. It must therefore be completed in full sentences and contain the exact wording you wish to be published.

2) If any authors received a salary from any of your funders, please state which authors and which funders..

9) Thank you for uploading your study's underlying data set. Unfortunately, the repository you have noted in your Data Availability statement does not qualify as an acceptable data repository according to PLOS's standards.

At this time, please upload the minimal data set necessary to replicate your study's findings to a stable, public repository (such as figshare or Dryad) and provide us with the relevant URLs, DOIs, or accession numbers that may be used to access these data. For a list of recommended repositories and additional information on PLOS standards for data deposition, please see: https://journals.plos.org/plospathogens/s/recommended-repositories

10) Please send a completed 'Competing Interests' statement, including any COIs declared by your co-authors. If you have no competing interests to declare, please state "The authors have declared that no competing interests exist". Otherwise please declare all competing interests beginning with the statement "I have read the journal's policy and the authors of this manuscript have the following competing interests".

**Reviewers' Comments:**

Reviewer's Responses to Questions

**Part I - Summary**

Reviewer #1: Sadri et al. present a novel sequencing method, HIV Single Molecule Real Time Capture (HIV SMRTcap), which combines long-read, single-molecule, real-time sequencing with a custom computational pipeline. The approach simultaneously identifies HIV integration sites and assesses proviral integrity at the single-molecule level in HIV-infected cell reservoirs. The manuscript shows that HIV SMRTcap performs effectively across multiple HIV subtypes – including A, B, C, D, and recombinant A/D – and is compatible with both blood and tissue samples, including from individuals on antiretroviral therapy with undetectable viral loads. Overall, the HIV SMRTcap method enables a detailed analysis of HIV reservoirs and will provide insights into HIV persistence of the virus in different compartments, contributing to efforts toward an HIV cure.

The HIV SMRTcap sequencing method is innovative and powerful. The method enriches fragmented genomic DNA containing integrated HIV using a pan-subtype oligonucleotide pool, enabling the detection and sequencing of multiple HIV subtypes while simultaneously identifying integration sites and proviral integrity. The manuscript presents thorough methodological validation using cells infected in vitro with different HIV subtypes, and presents analysis of sensitivity and of comparative performance relative to standard reservoir analysis techniques. The potential utility of the method is demonstrated by its use in characterizing HIV reservoirs in blood and tissue samples from individuals infected with diverse HIV subtypes. Overall, the technology represents a valuable tool for future research, particularly in studies involving under-characterized HIV subtypes.

Below are specific comments intended to help improve the manuscript for publication. Of note, would recommend page and line numbering for future submissions to facilitate review.

Reviewer #2: Please find the summary in my attached review.

Reviewer #3: Sadri et al describe the development and validation of "HIV SMRTcap" which is a novel method for characterizing HIV-1 reservoirs. The technique combines targeted oligonucleotide capture with long-read single-molecule real-time (SMRT) sequencing. Its major advantage is the simultaneous identification of proviral integration sites and the assessment of proviral integrity from a single DNA molecule, thereby addressing significant limitations of existing short-read or PCR-based methods. The authors demonstrate HIV SMRTcap across various HIV-1 subtypes, in different sample types including cell lines and primary tissues, and in challenging (low viral load) clinical samples from ART-suppressed individuals.

Overall, this is a well-written and technically solid study that introduces a powerful new tool for HIV research. The method is thoughtfully designed, the validation is largely comprehensive, and the data clearly demonstrate the potential of this approach, including in non-B subtypes and in low-viral-load.

I commend the authors for making their analysis pipeline available on GitHub and for depositing their sequencing data in the SRA. This commitment to open science will undoubtedly facilitate the adoption of their method.

**Part II – Major Issues: Key Experiments Required for Acceptance**

Reviewer #1: 1. Figure 2C: It seems there is one integration site detected by LM-PCR on the Y-chromosome. Yet, the parental cell line to generate 8e5 cells was derived from a female with acute lymphoblastic leukemia. How to explain this finding? Could this represent a technical/analysis artifact that has also affected other results in the study?

2. Figure 5B: The manuscript states “Robust HIV SMRTcap detection of >200 HIV templates was observed at the lower dilutions (e.g., containing more HIV templates) than expected.” This reviewer found this sentence to be confusing, and possibly inaccurate. Would it be fair to rewrite the sentence as “Robust HIV SMRTcap detection of >200 HIV templates was observed at lower dilutions than expected, suggesting greater sensitivity than expected.”? Also, the figure seems to suggest that more 8e5 cell DNA may have been loaded in the first dilution than intended, so that detecting HIV templates at the lower dilution does not necessarily reflect high assay sensitivity. Please clarify this.

3. Figure 5C: HIV copies recovered with SMRTcap were higher in the 10 HIV input template dilutions than in the 100 HIV input template dilutions. How to explain this discrepancy?

Reviewer #2: Please find key experiments in my attached review.

Reviewer #3: (No Response)

**Part III – Minor Issues: Editorial and Data Presentation Modifications**

Reviewer #1: 4. Table 1 shows the sequences used to design HIV specific probes, but does not indicate probe sequences or target genes. Please include a table indicating sequences and targets for each probe.

5. This reviewer could not find Supplementary Table 1.

6. Figure 2C: there are two open square symbols displayed on the outer chord chr 2 - are these integration sites displayed on the wrong chord?

7. Figure 3C: Text in the manuscript indicates that “93-RW/subtype A showed significantly more small deletions of HIV accessory genes vif, vpr, vpu, and nef than the other viruses,” yet the figure shows that deletion of nef in the subtype A virus was not significantly different compared to other subtypes. Please clarify this text.

8. Figure 4: Colors indicating internal deletion or truncated proviruses are difficult to distinguish on the chord diagram, please consider changing color selections.

9. Figure 5A: This reviewer was unclear on what the horizontal dashed line represents.

10. Figure 6: How many PBMC were processed to characterize the reservoir in ART-treated participants P4, Donor 17 and Donor 13? For Donor 17 and Donor 13? Are the differences in the number of proviruses recovered in each timepoint related to the input of cells for sequencing?

11. Figure 6B: Two clones are shown integrated in chromosome 17 at positions only 5 bases apart (40646538 and 40646543). The manuscript suggests this likely represents the same clone captured from either the 5' or 3' end of the integrated provirus. Please clarify whether a similar situation applies to the two clones integrated in chromosome 5. Additionally, the boxes indicating integration sites currently overlap with the pie chart, obscuring the dots representing individual proviruses. It would be best to reposition these boxes so that all data points remain visible.

12. Figure 6: The plots from all donors show that there are some clones with the same integration site, but it is unclear if these are the same clone sheared in different locations or if instead they are different clones that differ by a few bases. Please include a table with all information of the integration sites (Ex: chromosome, location, and provirus orientation).

Reviewer #2: Please find minor issues in my attached review.

Reviewer #3: 1. key conclusion of the paper is that HIV SMRTcap provides a more detailed view of proviral integrity than qPCR-based methods like the Intact Proviral DNA Assay (IPDA): SMRTcap identifies accessory gene deletions that IPDA would miss (Fig 7B-D). While technically true, the discussion of this point requires more nuance.

The IPDA is designed to quantify proviruses that are likely replication-competent, using probes in gag and env as a proxy. HIV SMRTcap, by contrast, defines "intactness" bioinformatically as having >75% of all 11 viral genes present. It is not clear that the accessory gene deletions identified exclusively by SMRTcap would render a provirus replication-incompetent. Therefore, calling these proviruses "defective" in the same category as those with major structural gene deletions might be an overstatement. The authors should discuss this distinction more carefully. Are these assays measuring different things (potential for replication vs. genomic completeness), and if so, how should they be used together? The authors are probably right, but neither approach is the “gold standard”, i.e., actual replication assays.

The 75% threshold for calling a gene "intact" feels somewhat arbitrary. Some justification for how this cutoff was determined and validated would strengthen the integrity classification schema. How sensitive are the results to this specific value?

2. The limit of detection experiment is useful and informative, but the results presented in Figure 5 raise questions. The authors show that at very low dilutions, the empirically detected number of HIV templates exceeds the number predicted by their subsampling model. This is an unexpected and interesting result.

Could the authors speculate on why this might be? Does the model, which is based on technical replicates, not fully capture the dynamics of the assay? Could there be more efficient hybridization or capture when the target is extremely rare? A more thorough discussion of this discrepancy would be beneficial.

The variability in detection at the lowest dilution (2 of 4 replicates) should also be acknowledged more directly in the discussion of the assay's sensitivity. While impressive, users need to be aware of the stochastic nature of detection at such low frequencies.

3. The use of unique shearing-site "endogenous barcodes" to distinguish PCR duplicates from true clonal expansion is clever. However, the operational definition of an identical clone—integration sites falling within a 10 bp window—could be problematic. While intended to account for polymerase slippage, it also risks conflating distinct integration events in genomic "hotspots" into a single clone. Providing a reference or further justification for this 10 bp window would be helpful. Furthermore, the authors correctly identify a current limitation in Figure 6, where 5' and 3' single-flanked reads from what could be the same clone are analyzed separately. Because could lead to an underestimation of clone size and an overestimation of clonal diversity, this limitation should be more prominently stated in the Results and Discussion sections, rather than being presented primarily as a future pipeline development.

More minor issues

1 In the clonality pie charts (Fig 6D, 6F), it is difficult to track the clinically relevant clones mentioned in the text (e.g., in CCR3 or BACH2) without any labels. Annotating the largest or most persistent clones directly on the charts or in a corresponding table would greatly aid interpretation.

2. The circos plot in Figure 2B contains a lot of information (genic/intergenic, repeat/non-repeat). Distinguishing between the triangles and circles, and open versus filled symbols, is difficult at a glance. These figures (and some others) also violate the cardinal Tufte data viz rule, by having low data to ink ratio (a lot of wasted white space for example).

3. In the discussion, the authors note that HIV SMRTcap likely fails to capture hypermutated proviruses, a known limitation of hybridization-based methods. They reasonably argue this is "not considered a critical fault" since such viruses are _probably_ not replication-competent. However, for studies aiming to characterize the entire landscape of proviral DNA, this is a notable bias that should be clearly and strongly stated.

4. The discussion claims the method is "cost-effective." While this may be true, the manuscript provides no explicit data to support this. I would suggest either removing this statement or softening it (e.g., "HIV SMRTcap may reduce overall costs by consolidating multiple assays...").

PLOS authors have the option to publish the peer review history of their article (what does this mean? ). If published, this will include your full peer review and any attached files.

**Do you want your identity to be public for this peer review?** For information about this choice, including consent withdrawal, please see our Privacy Policy .

Reviewer #1: No

Reviewer #2: No

Reviewer #3: No

**Figure resubmission:**

**Reproducibility:**



---

## [Decision Letter · Decision Letter 1]

31 Oct 2025

PPATHOGENS-D-25-00898R1

Development and validation of HIV SMRTcap for the characterization of HIV-1 reservoirs across tissues and subtypes

PLOS Pathogens

Dear Dr. Smith,

Thank you for submitting your manuscript to PLOS Pathogens. After careful consideration, we feel that it has merit but does not fully meet PLOS Pathogens's publication criteria as it currently stands. Therefore, we invite you to submit a revised version of the manuscript that addresses the points raised during the review process.

Please submit your revised manuscript within 30 days Dec 30 2025 11:59PM. If you will need more time than this to complete your revisions, please reply to this message or contact the journal office at plospathogens@plos.org. Please include the following items when submitting your revised manuscript:

We look forward to receiving your revised manuscript.

Kind regards,

Jason M. Brenchley

Academic Editor

PLOS Pathogens

Susan Ross

Section Editor

PLOS Pathogens

Sumita Bhaduri-McIntosh

Editor-in-Chief

PLOS Pathogens

orcid.org/0000-0003-2946-9497

Michael Malim

Editor-in-Chief

PLOS Pathogens

orcid.org/0000-0002-7699-2064

**Additional Editor Comments:**

The reviewers are all impressed with the work and are satisfied that the substance of their concerns have been addressed. One reviewer raises several important minor points which need to be addressed.

**Reviewers' Comments:**

Reviewer's Responses to Questions

**Part I - Summary**

Reviewer #1: No further concerns, recommend publication.

Reviewer #2: The authors have incorporated new data and analyses in this revised manuscript, as well as editing text and figures for improved clarity, which have appropriately responded to reviewer critiques and improved the paper. However, there are remaining minor errors/typos which need correction for the manuscript to be ready for publication. Please address these minor revisions listed below.

Reviewer #3: The authors have diligently addressed all the comments from the previous round of reviews. The manuscript is now suitable tor publication.

**Part II – Major Issues: Key Experiments Required for Acceptance**

Reviewer #1: (No Response)

Reviewer #2: (No Response)

Reviewer #3: (No Response)

**Part III – Minor Issues: Editorial and Data Presentation Modifications**

Reviewer #1: (No Response)

Reviewer #2: 1. Figure 7 is now an independent figure, while it was previously part of Figure 6, but there is no figure legend for the new Figure 7. Please align the figure legends and figures (move current Figure 6E-F legend to independent Figure 7 legend, and make current Figure 7 legend the Figure 8 legend).

2. There appears to be a missing arrow for the BACH2 integration clone for Figure 7B, Y9. The arrow is also missing in Figure 2B for chromosome 13 integration.

3. The significance asterisks are missing in Figure 3C.

4. Newly included QVOA data in tables for Figures 6 and 7 have not been cited for the source of the data.

5. It seems a library of p88 of 8e5 cells was used for SMRTcap sequencing and comparison to LM-PCR in Figure 2C. If so, please include the sequencing read data in Table 3 and in a GenBank BioProject.

6. Figure 6A contains no Y chromosome in the Circos plot although these data are from the male patient P4.

7. It appears that in response to reviewer comments, what were originally two proviruses integrated in chromosome 5 in Figure 6A were re-analyzed and designated as a single provirus/clone. This makes 6 proviruses, which are shown in Figure 6A and 6B, but text in line 345 of the manuscript clean copy states there are 7. Also, the gene given in line 350 (ENSG00000264058) does not match the gene in Figure 6B.

8. Line 196 should direct to Table 3, and line 540 should direct to Table 2.

9. Table 1 still appears to contain the typo for the ID of SE365, which should be AB485649.

10. The methods for PDX mice should state 13 weeks post-engraftment or eight weeks post-infection in line 572, and inclusion of dose and route of HIV infection information in this section of the methods would be appropriate.

11. Typos in line 538 (should be Alemtuzumab), 366 (should be ENSG00000288724 per figure 6D), and line 245 (between 0% and 75%); additional typos in lines 755 and 1075.

Reviewer #3: (No Response)

PLOS authors have the option to publish the peer review history of their article (what does this mean? ). If published, this will include your full peer review and any attached files.

**Do you want your identity to be public for this peer review?** For information about this choice, including consent withdrawal, please see our Privacy Policy .

Reviewer #1: No

Reviewer #2: No

Reviewer #3: No

**Figure resubmission:**
---

## [Editor Report · Decision Letter 2]

16 Dec 2025

Dear Dr Smith,

We are pleased to inform you that your manuscript 'Development and validation of HIV SMRTcap for the characterization of HIV-1 reservoirs across tissues and subtypes' has been provisionally accepted for publication in PLOS Pathogens.

Best regards,

Jason M. Brenchley

Academic Editor

PLOS Pathogens

Susan Ross

Section Editor

PLOS Pathogens

Sumita Bhaduri-McIntosh

Editor-in-Chief

PLOS Pathogens

orcid.org/0000-0003-2946-9497

Michael Malim

Editor-in-Chief

PLOS Pathogens

orcid.org/0000-0002-7699-2064

The authors have addressed the minor comments raised by one reviewer.
---

## [Editor Report · Acceptance letter]

Dear Dr Smith,

We are delighted to inform you that your manuscript, "Development and validation of HIV SMRTcap for the characterization of HIV-1 reservoirs across tissues and subtypes," has been formally accepted for publication in PLOS Pathogens.

Best regards,

Sumita Bhaduri-McIntosh

Editor-in-Chief

PLOS Pathogens

orcid.org/0000-0003-2946-9497

Michael Malim

Editor-in-Chief

PLOS Pathogens

orcid.org/0000-0002-7699-2064